# A quaternary tetramer assembly inhibits the deubiquitinating activity of USP25

Bing Liu [1,2], Marta Sureda-Gómez [3], Yang Zhen[1,2], Virginia Amador[3] & David Reverter[1,2]

USP25 deubiquitinating enzyme is a key member of the ubiquitin system, which acts as a positive regulator of the Wnt/β-catenin signaling by promoting the deubiquitination and stabilization of tankyrases. USP25 is characterized by the presence of a long insertion in the middle of the conserved catalytic domain. The crystal structure of USP25 displays an unexpected homotetrameric quaternary assembly that is directly involved in the inhibition of its enzymatic activity. The tetramer is assembled by the association of two dimers and includes contacts between the coiled-coil insertion domain and the ubiquitin-binding pocket at the catalytic domain, revealing a distinctive autoinhibitory mechanism. Biochemical and kinetic assays with dimer, tetramer and truncation constructs of USP25 support this mechanism, displaying higher catalytic activity in the dimer assembly. Moreover, the high stabilization of tankyrases in cultured cells by ectopic expression of a constitutive dimer of USP25 supports a biological relevance of this tetramerization/inhibition mechanism.

[1] Institut de Biotecnologia i de Biomedicina (IBB), Universitat Autònoma de Barcelona, Bellaterra 08193, Spain. [2] Dept. de Bioquímica i Biologia Molecular, Serra Hunter Fellow, Universitat Autònoma de Barcelona, Bellaterra 08193, Spain. [3] Institut de Investigacions Biomèdiques Agustí Pi i Sunyer (IDIBABS), Barcelona 08036, Spain. Correspondence and requests for materials should be addressed to D.R. (email: david.reverter@uab.cat)

The ubiquitin system is a universal means of protein regulation which controls a wide range of cellular processes[1–3]. The conjugation of ubiquitin to target proteins is conducted via a conserved multistep enzymatic cascade through E1, E2, and E3 enzymes, resulting in the formation of an isopeptide bond between the C-terminus of ubiquitin and the lysine on the target protein[4]. Reversely, deubiquitinating enzymes (DUBs) can remove ubiquitin by catalyzing the hydrolysis of the isopeptide bond. Therefore, ubiquitin conjugation and deconjugation are balanced and are tightly regulated by E3 ligases and DUBs. There are more than 100 DUBs encoded in the human genome[5]. Based on the structure studies, their isopeptidase activities are conferred by distinct catalytic domains[5–7]. Recently, the identification of a new family MINDY extends the DUBs to six distinct families[8]. In human genome, ubiquitin-specific proteases (USPs) constitute the largest family of DUBs counting more than 50 members and playing important roles in a wide variety of cellular processes[5–7,9].

Structural studies show that USPs have a common conserved fold consisting of three subdomains known as Palm, Thumb, and Fingers, in which the active site cysteine is located between the *Palm* and *Thumb*, while the *Fingers* grip the "distal" ubiquitin[10]. However, their catalytic activities are usually modulated through their different additional domains. For example, a C-terminal 19 amino acid peptide binds the activation cleft in the catalytic domain and stabilizes the catalytically competent conformation, thus enhancing the activity of USP7[11]. This activation can be enhanced allosterically by the metabolic enzyme GMPS[12]. The binding of USP4 N-terminal DUSP-Ubl domain promotes a change of a switching loop near the active site, and hence enhances ubiquitin dissociation and makes it achieve full catalytic activity[13].

USP25 shows activity in hydrolyzing K48- and K63-linked polyubiquitin chains[14,15] and is a target for different post-translational modifications including phosphorylation, SUMOylation, and ubiquitination. The SYK non-receptor tyrosine kinase can specifically phosphorylate USP25 and decrease its cellular levels[16]. USP25 enzymatic activity is suppressed after vaccinia-related kinase 2-mediated phosphorylation at Thr680, Thr727, and Ser745, which controls the stability of the eukaryotic chaperonin TRIC[17]. SUMO and ubiquitin regulate the catalytic activity of USP25 by conjugation at Lys99 with opposite effects: ubiquitination activates while SUMOylation inhibits the USP25 activity[15,18].

USP25 has been associated with inflammation, immune response, and cancer. For example, it is associated with endoplasmic reticulum-associated degradation[19], and acute endoplasmic reticulum (ER) stress regulates amyloid precursor protein processing through ubiquitin-dependent degradation by USP25[20]. USP25 is a negative regulator of interleukin-17-mediated signaling and inflammation through the removal of ubiquitination in TRAF3, TRAF5, and TRAF6 and can regulate TLR4-dependent innate immune responses[21–23]. Moreover the type I interferon-IRF7 can activate the expression of USP25 gene, which is essential for innate immune signaling[24]. Several evidences point to the involvement of USP25 in cancer. USP25 gene is found greater than threefold overexpression in human breast cancer tissue[25], and further study shows that it as a putative tumor suppressor in human lung cancer[26]. Another study in human non-small-cell lung cancer reveals that miR-200c inhibits invasion and metastasis by directly targeting USP25 and reducing the expression level[27]. Recently, it has been shown that USP25 directly interacts with tankyrases through its C-terminal tail and promotes their deubiquitination and stabilization, thus regulating Wnt/β-catenin signaling pathway, making an important impact in cell proliferation and human cancer development[28].

Despite these roles in cellular pathways, structural insights of USP25 activity regulation are still not clear. The N-terminal domain (NTD) of USP25 contains one ubiquitin-associated domain, one SUMO-interacting motif, and two ubiquitin-interacting motifs (UIM1 and UIM2)[29]. SUMO-modified USP25 maintains the ability to hydrolyze Ub-AMC, but exhibits lower activity during polyubiquitin chain degradation, which indicates that SUMO modification affects the recruitment of ubiquitin but not the catalytic domain activity[15]. Although the basic elements of the catalytic domain of USP25 are conserved, there is a large domain (175 residues) inserted within the catalytic core. Also, previous studies show that USP25 can form homodimers or oligomers in vivo and the catalytic domain (between residues 153 and 679) is relevant for this dimerization or oligomerization[15,18]. Finally, the C-terminal domain (CTD) of USP25 is involved in phosphorylation and substrate recognition[16,17], directly interacting with tankyrases to promote their deubiquitination and stabilization[28].

In this study, we present the crystal structure of USP25 catalytic domain. Strikingly, the USP25 structure displays a homotetramer quaternary assembly and the catalytic domain can be divided in three differentiated subdomains: a general palm-like catalytic domain (USP-like), a long coiled-coil (LCC), and an "inhibitory loop" (IL-loop or IL). Whereas the USP-like catalytic domain is similar with the other USPs, the LCC and IL-loop domains are unique to USP25. In our USP25 structure, each IL-loop inserts into the catalytic domain from another molecule of the tetramer, thus preventing substrate binding and thereby impairing USP25 activity. Moreover, this tetramerization/inhibition mechanism of USP25 has been analyzed in a cellular context with the stabilization of tankyrases, which directly regulate the Wnt/β-catenin signaling pathway. This mechanism for USP25 activity regulation has not been described for other deubiquitinases and probably unveils a paradigmatic type of regulation in the USP family[30].

## Results

**USP25 recombinant expression produces two oligomer forms**. Human USP25 is a modular protein composed by three domains: an NTD (Met1 to Tyr159), a central USP-like domain (Asp160 to Glu714), and a CTD (Lys715 to Arg1055) (Fig. 1a and Supplementary Figure 1). Since the CTD does not directly participate in the catalytic activity and the expression levels of the full-length USP25 in *Escherichia coli* were low[14], we produced a USP25 construct (USP25NCD) including the NTD and the USP-like catalytic domain (from Gln18 to Glu714).

A particular feature of the recombinant expression of human USP25 in *E. coli* is the presence of two different stable oligomeric states, dimer and tetramer, which are stable after running two different size-exclusion and one ionic exchange chromatographies (Fig. 1b). USP25 oligomerization has also been reported by two groups in human cell lines using a dual-tag expression system[15,18], but the type of the quaternary assembly could not be determined. Mass spectroscopy analysis of recombinant USP25 did not reveal any particular feature, such as post-translational modifications, to explain the presence of these stable oligomer fractions.

However, activity assays using the standard fluorescence substrate for deubiquitinating activity, Ub-AMC, revealed marked differences between dimer and tetramer, with the dimer assembly of USP25 being substantially more active than the tetramer (Fig. 1c). These activity differences are maintained despite the presence of cross-contamination between oligomers, as observed in non-denaturing polyacrylamide gel electrophoresis (PAGE) analysis (Fig. 1c). To get more insight into these activities, we

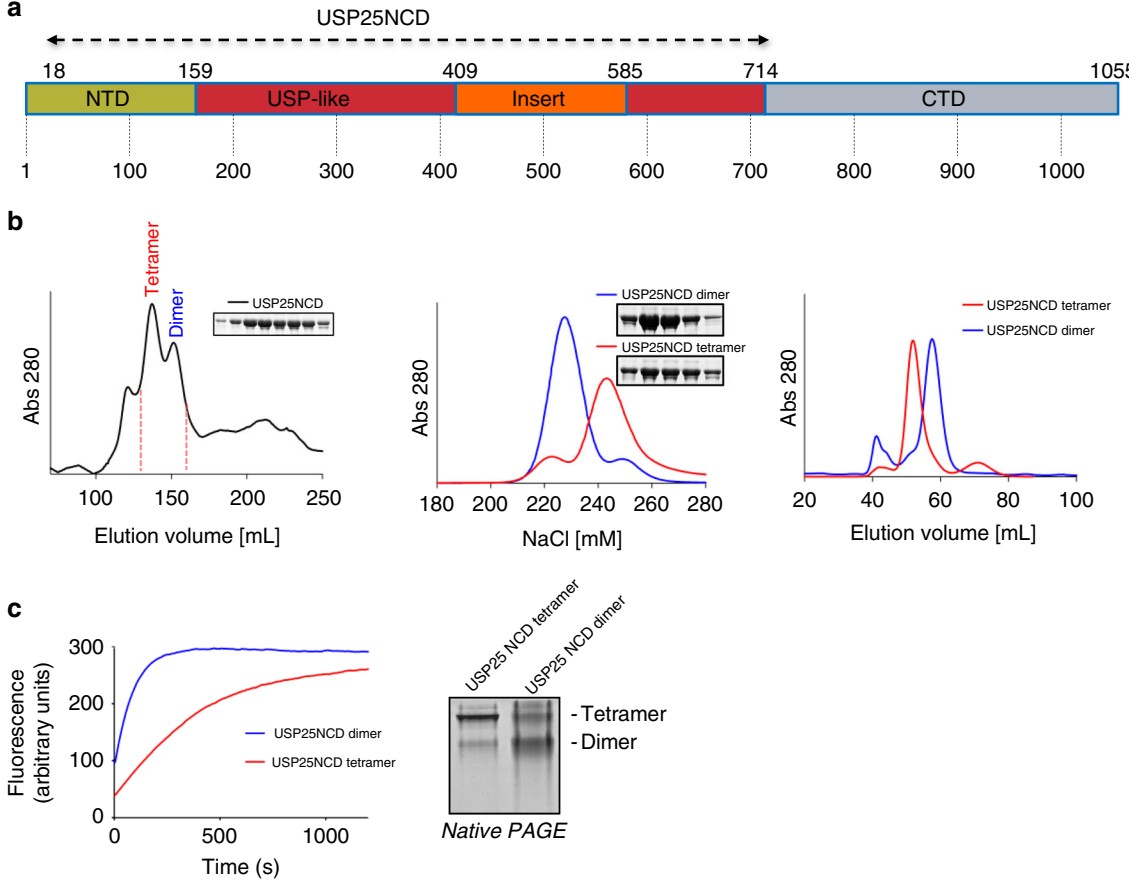

**Fig. 1** Recombinant USP25 is produced as a tetramer or a dimer in *E. coli*. **a** Scheme of the USP25 protein domains. The construct for crystallization is from residues 18 to 714 (USP25NCD). **b** First preparative gel filtration purification of USP25 in Superdex 200 column, inset shows the SDS-PAGE of the indicated fractions. Anionic-exchange purification of the two peak fractions of the previous gel filtration, inset shows the SDS-PAGE of the dimer and tetramer fractions. Analytical gel filtration purification of the dimer and tetramer fractions of USP25 in Superdex 16 column. **c** Plot of the deubiquitinating activity assays of dimer and tetramer with Ub-AMC as a substrate. Non-denaturing native PAGE of the dimer and tetramer fractions of USP25NCD. All uncropped gels and blots are displayed in Supplementary Figure 12

conducted crystallization trials with these two oligomeric assemblies of USP25.

**The crystal structure of USP25.** Interestingly, only the tetramer assembly of USP25 gave rise to single crystals. Initial diffraction of the USP25 crystals was poor, never beyond 6 Å resolution; however, after conducting a lysine methylation protocol[31], diffraction was notably improved, and thus finally collecting a 3.28 Å resolution dataset. Apparently, the lysine methylation reaction did not affect the deubiquitinating activity of USP25 (Supplementary Figure 2b, c). The crystals belonged to the *I*422 space group, and contained one molecule per asymmetric unit (Table 1). The USP25 structure was solved by SAD using mercurial derivatives, three of which were located bound to three cysteine residues (one corresponding to the active site cysteine). After several steps of modeling and refinement using the structure of USP7 as a guide, the electron density maps clearly showed the elements of the catalytic USP-like domain (general "palm-like" structure), plus a long helical coiled-coil domain emanating from the middle of the USP-like domain (Fig. 2a–c), which corresponded to the long sequence insertion splitting the catalytic domain. The final electron density map model includes most of the chain for the catalytic domain (Tyr159 to Ala708) with a notable absence of the whole NTD (Gly18 to Pro158), and of some disordered loops connecting secondary structure elements

in the USP-like domain (Asn205 to Lys214; Asn256 to Gln260; Ile346 to Ser354; Asp471 to Arg515; Thr671 to Gly677).

**Structure of the USP-like catalytic domain of USP25.** The USP-like domain of USP25 contains all characteristic elements of the USP family, which was initially compared to a "right hand" that could entrap ubiquitin[10], including Palm, Fingers, and Thumb subdomains (Fig. 2c). The distances between the catalytic triad residues indicate that the active site might be preformed in the absence of the ubiquitin substrate, with distances ranging around 3.2 Å between the sulfur of Cys178 and the imidazole ring of His607, and 3.0 Å between His607 and Asn624 (Fig. 2d and Supplementary Figure 3), which is comparable to the USP7-Ub aldehyde complex structure (rmsd 1,67 Å for 268 aligned residues)[10]. The preformed active site in USP25 differs from other apo structures of USPs, in which the catalytic triad residues are too far for catalysis and binding of the "distal" ubiquitin substrate is required to rearrange the catalytic triad to an active conformation (e.g., USP7, USP14, USP18)[11,32,33].

Unexpectedly, the long sequence insertion of the USP25 catalytic domain (between Asn408 and Met586) forms a coiled-coil structure (LCC) composed by two long α-helices, which we named helix $LCC_1$ (from Arg409 to Ser460) and helix $LCC_2$ (from Leu540 to Ser585) (Fig. 2b, e). It is interesting to observe a break in the α-helix $LCC_1$ produced by a "kink" between Ser438

                    3

**Table 1 Data collection and structure refinement statistics**

| Data collection | | |
|---|---|---|
| Beamline | ALBA-XALOC | ALBA-XALOC (Hg²⁺) |
| Space group | *I*422 | *I*422 |
| Wavelength (Å) | 0.9791 | 0.9998 |
| Resolution (Å) | 95.08–3.28 (3.29–3.28)ᵃ | 94.76–4.35 (4.35–4.37) |
| $a, b, c$ (Å) | 140.806, 140.806, 190.156 | 141.17,141.17,190,17 |
| $\alpha, \beta, \gamma$ (°) | 90, 90, 90 | 90, 90, 90 |
| Unique reflections | 14,883 | 6588 |
| Data redundancy | 6.50 (6.40) | 25.4 (26.5) |
| $R_{merge}$ | 0.07 (0.98) | 0.19 (1.06) |
| CC (1/2) | 0.99 (0.85) | 0.99 (0.97) |
| $I/\sigma$ | 16.3 (2.3) | 20.3 (6.9) |
| Completeness (%) | 99.5 (97.7) | 99.9 (95.7) |
| Refinement | | |
| Resolution (Å) | 95.08–3.28 (3.37–3.28) | |
| Non-anomalous reflections | 14,150 | |
| $R_{work}/R_{free}$ | 0.202/0.275 | |
| Number of all atoms | 3963 | |
| *B*-factor protein (Å²) | 129.6 | |
| RMSD bond (Å)/ angle (°) | 0.011/1.524 | |
| Ramachandran plot | | |
| Favored (%) | 91.99 | |
| Allowed (%) | 6.71 | |
| Disallowed (%) | 1.30 | |

ᵃ Values in parentheses are for the highest-resolution shell

and Pro447 (Fig. 2e), which is involved in the assembly of the tetramer structure (as will be explained later). The two α-helices of the coiled coil are connected by a long loop, which we named IL-loop based on its role in the regulation of the enzymatic activity of USP25. The IL-loop is composed by a non-conserved sequence (Fig. 2g), absent in the USP25 crystal structure (from Asp471 to Arg515), and by a conserved sequence, which can be clearly observed in the electron density maps engaging contacts with the catalytic domain of a different molecule of the crystal. This interaction is essential for the tetramer assembly.

**USP25 tetramer is formed by the interaction of two dimers**. The structure of USP25 reveals the presence of a homotetramer quaternary assembly composed by the interaction of four different USP25 molecules from the crystal lattice (Fig. 3). PISA server analysis also predicts that the tetramer structure of USP25 (A₄), with a buried surface area of 23,730 Å², is formed by the assembly of two homodimers (2 × A₂) (Fig. 3). Based on the high number of contacts, the dimer assembly might represent a minimal stable oligomer state of USP25. The structure of USP25 can thus explain the presence of the tetramer and dimer fractions observed during the purification by gel filtration (Fig. 1).

Each homodimer composing the USP25 tetramer is comprised by an extended contact interface that includes approximately half of the LCC coiled-coil domain, forming a four-helix bundle motif, which would act as a dimerization hub between two USP-like catalytic domains, resembling a "pair of cherries" with the knot at the end of their stems (Figs. 2f and 3). PISA server analysis reveals a 2110 Å² dimer interface involving a total of 55 residues. Most residues are hydrophobic (e.g.,. Leu448, Leu452, Phe458, Leu548, Trp555) (Supplementary Figure 4a), but the dimer interface also includes 20 and 12 hydrogen bond and salt bridge

contacts, respectively (see contact list in Supplementary Figure 5). All contacts are highly conserved across species (Fig. 2g and Supplementary Figure 1a). The dimer structure exposes two USP-like catalytic domains to the solvent.

**IL-loop contacts the catalytic domain in the tetramer**. The LCC-IL-loop insertion domain also participates in the tetramer interface, but in contrast to the dimer, the interaction occurs with the catalytic domain of the opposite dimer (Fig. 3). Two contact areas can be distinguished in the tetramer interface: between the extended IL-loop which is deeply inserted into the S1 ubiquitin-binding surface of the catalytic domain (Fig. 4a, c in green and Supplementary Figure 4c); and between the LCC₁ "kink" motif with residues from the loop connecting β9–β10 strands of the catalytic domain (Fig. 4a in blue and Supplementary Figure 4b). So, the interaction of each IL-loop with the catalytic domain of the opposite dimer is essential for the formation of a stable tetramer assembly.

The IL-loop was unambiguously observed in the electron density maps in contact with the S1 ubiquitin-binding pocket of the USP-like catalytic domain (Fig. 4b). Structural comparison with the USP7-ubiquitin complex (PDB 5JTJ) and with the CYLD-diubiquitin K63 complex (PDB 3WXG)[34] indicates that the IL-loop and the "distal" ubiquitin share a similar binding surface (Fig. 4b and Supplementary Figure 3), thus preventing substrate binding in the tetramer assembly. Such "tetramer-autoinhibition" mechanism would explain the different activities observed between the purified tetramer and dimer of USP25 (Fig. 1c).

The residues of the IL-loop involved in the tetramer interface, from Ile518 to Leu531, are mostly conserved across species in contrast to the rest of the IL-loop, not observed in the electron density maps (Fig. 2g). Among the high number of contacts conducted by the IL-loop, Pro521 and Phe522 are completely buried in a deep crevice formed between the α-helices α6, α10 and the central β-sheet of the USP-like catalytic domain, establishing several hydrophobic interactions: Val183 and Leu187 from α-helix α6; Leu271 from α-helix α10; Phe299 and Tyr300 from α-helix α11; and Met653 from β-sheet β16 of the USP25 catalytic domain (Fig. 4c, d and Supplementary Figure 4c and 5).

To assess the role of this interface, several USP25 point mutants were generated in the IL-loop and in the S1 ubiquitin-binding surface (Fig. 4e and Supplementary Figure 6). P521S and F522G IL-loop point mutants, which are involved in hydrophobic contacts within the pocket, disrupt the tetramer assembly and their deubiquitinating activity is comparable to the USP25 dimer. Similar results are observed for P528G, P535L, and S525P IL-loop point mutants, but in this instance the distorted conformation of the IL-loop probably perturbs the binding affinity. On the USP-like catalytic domain surface, both E373A and Q322A point mutants of the S1 ubiquitin-pocket disrupt three hydrogen bonds with the IL-loop and compromise the tetramer assembly, in particular E373A displays similar deubiquitinating activities as USP25 dimer. Finally, C651F and L271W point mutants were intended to lock the USP-like binding cleft, preventing the IL-loop binding. Whereas this was partially achieved by C651F, the L271W point mutant unexpectedly stabilized the tetramer with a consequent loss of deubiquitinating activity (Fig. 4e and Supplementary Figure 6).

Finally, to check a potential competition between the IL-loop and the ubiquitin substrate, USP25-deubiquitinating activities were measured in the presence of a peptide derived from the IL-loop binding sequence (HKPFTQSRIPPD) (Supplementary Figure 6e). The absence of inhibition produced by the IL-loop

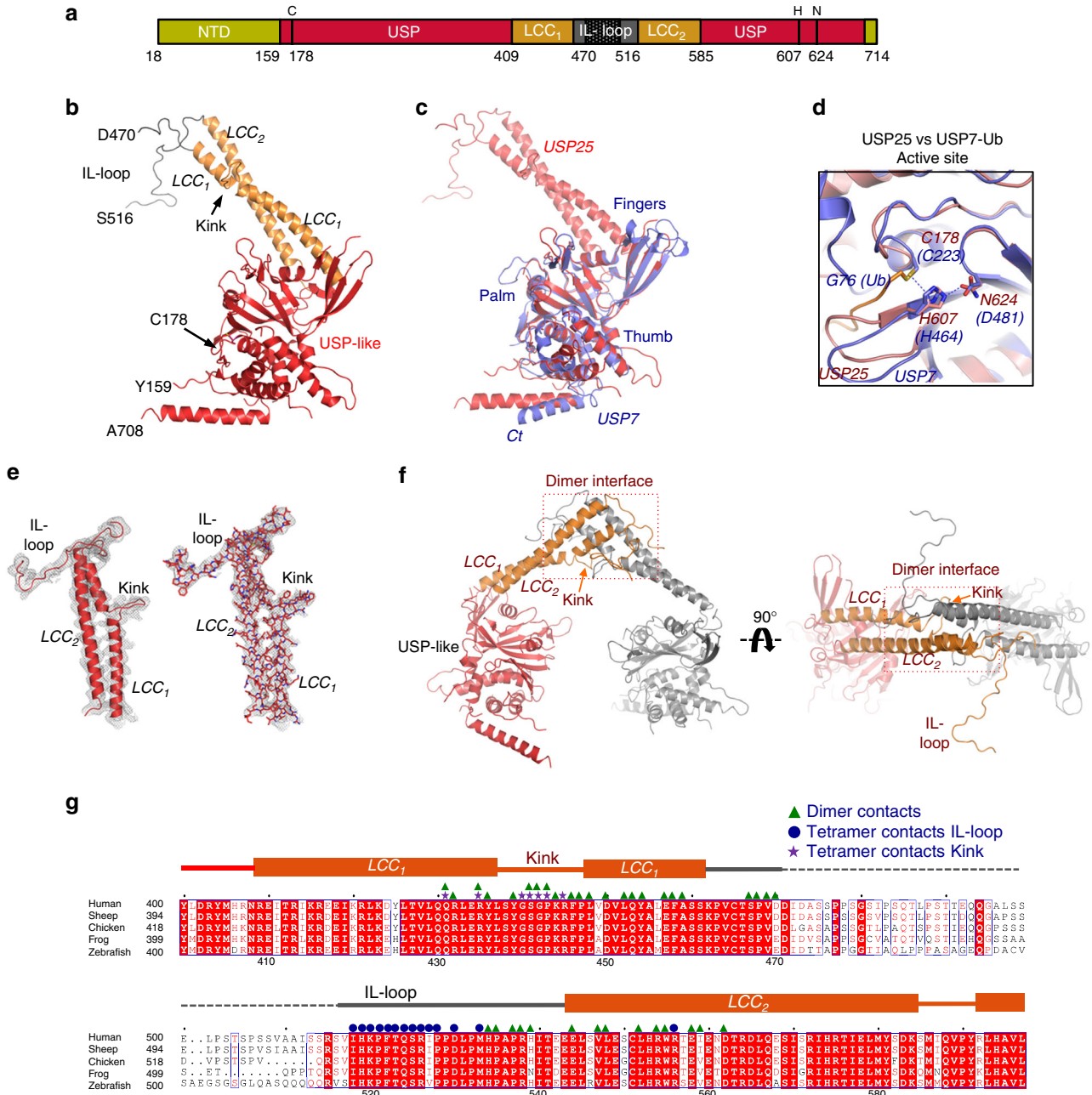

**Fig. 2** The crystal structure of USP25. **a** Scheme of the domain composition of the crystal USP25 construct. N-terminal domain (NTD) (18–158), and the non-conserved sequence (dashed gray color, 471–515) of inhibitory loop (IL-loop) are not observed in the final structure. **b** Ribbon representation of the different domains of the USP25 monomer. Red color shows the conserved USP-like domain, orange color the coiled-coil domain (LCC), and black color the domain IL-loop. Active site cysteine and domain boundaries are labeled. **c** Structural superposition between USP25 (red) and USP7 (blue) structures (PDB code 5JTJ). Palm-like domains are labeled. **d** Close-up view of the superimposition of active sites of USP25 and USP7. USP25 and USP7-ubiquitin aldehyde are shown in orange and blue, respectively. Catalytic triad residues from USP25 are labeled and shown in stick representation, whereas the corresponding residues from USP7 are shown within parentheses. Hydrogen bonds are represented by dashed lines. **e** Two representations of the electron density maps of the LLC and IL-loop domains of USP25. **f** Ribbon representation of the two monomers composing the dimer structure observed in the crystal of USP25 in two different orientations. **g** Sequence alignment of the LCC-IL-loop domain of USP25 across species. Green triangles indicate contacts in the dimer interface. Blue circles and purple stars indicate tetramer contacts by the IL-loop or the Kink region, respectively. All sequences are aligned online with Clustal Omega and formatted using ESPript[44]

peptide suggests a model in which the structural context in the tetramer assembly is required for the proper binding and thus inhibition.

Overall, this reciprocal point mutant analysis of the binding interface verifies the role of the IL-loop in the tetramer assembly and in the regulation of the deubiquitinating activity.

**USP25 truncation reveals the presence of dimer and tetramer.** The role of the LCC-IL-loop domain in oligomerization was assessed by two truncation constructs of USP25: IL-loop truncation (ΔIL, deletion from residue Cys465 to residue Pro537) and LLC-IL-loop truncation (ΔLCC-IL, deletion from residue Arg416 to residue Leu579). These different constructs were

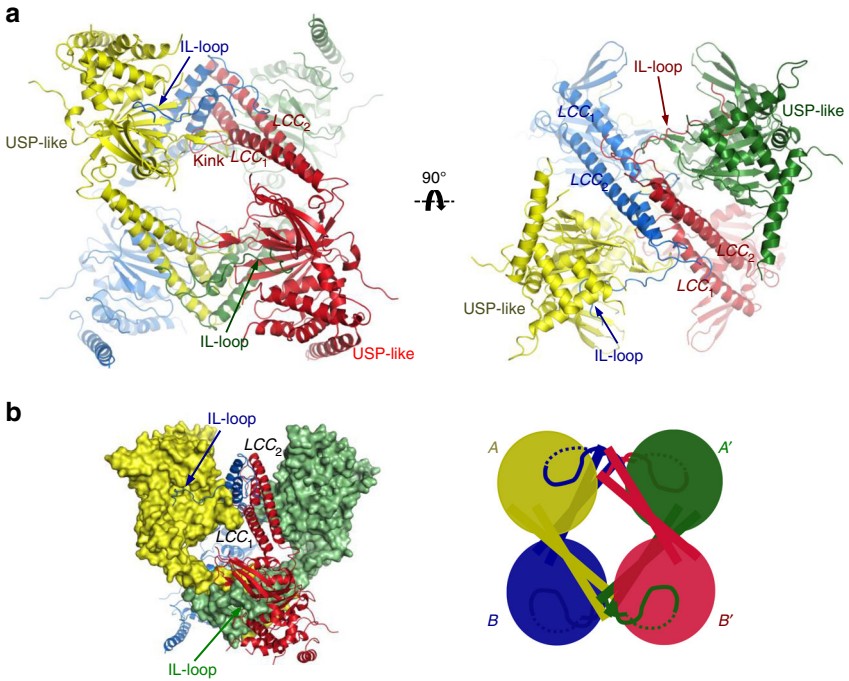

**Fig. 3** USP25 is a homotetramer formed by the assembly of two dimers. **a** Ribbon representation of the USP25 tetramer. Each subunit is represented with a different color. LLC and IL-loop domains are labeled. Different orientations of the USP25 tetramer. **b** Mixed surface and ribbon representation of the homotetramer structure of USP25. Cartoon image of the tetramer assembly formed by the assembly of two homodimers (A, A' homodimer and B, B' homodimer)

checked in the context of the full-length of USP25 (USP25FL), the CTD deletion (USP25NCD) and the NTD and CTD deletion (USP25CD) (see scheme in Fig. 5a). All USP25 truncations did not display expression problems and were eluted as single peaks during their purification by gel filtration chromatography.

All USP25 truncation were run in parallel through an analytical size-exclusion chromatography under identical experimental conditions, which permitted the correlation between the expected molecular weight and the gel filtration elution times (Fig. 5b). Interestingly, full-length USP25FL, USP25NCD, and USP25CD elution times were compatible with the presence of a tetramer structure, whereas deletion of the IL-loop (ΔIL) and deletion of the LLC-IL-loop (ΔLCC-IL) resulted in elution times compatible with dimer and monomer structures, respectively (Fig. 5b). Additionally, dynamic light scattering (DLS) analysis of the USP25 truncation constructs also displayed a good correlation between oligomers and their theoretical molecular weight (Supplementary Figure 7). These experiments confirm the role of the LLC coiled-coil domain in the formation of the dimer assembly, and the role of the IL-loop in the formation of the tetramer assembly.

**USP25 dimers display higher deubiquitinating activities**. Interestingly, all dimer constructs (ΔIL USP25FL, USP25NCD and USP25CD) displayed a higher hydrolysis of the Ub-AMC substrate in comparison to all tetramer (USP25FL, USP25NCD, and USP25CD) and monomer (ΔLCC-IL USP25FL, USP25NCD, and USP25CD) constructs (Fig. 5c). We also compared the purified dimer of USP25 (Fig. 1) and the truncation dimer construct, ΔIL USP25NCD (Fig. 5c), displaying both comparable deubiquitinating activities and similar purification profiles and native PAGE migrations (Supplementary Figure 7b). The higher activity of the dimer over the tetramer was also confirmed using polyubiquitin chains, either K48- or K63-linked, and diubiquitin K48-linked substrates (Supplementary Figure 8).

Quantitative activity analyses were also conducted by steady-state "Michaelis–Menten" kinetics with the purified tetramer and the "constitutive" truncated dimer and monomer of USP25 (Fig. 5d and Table 2). The higher $K_{cat}/K_M$ values displayed by the dimer $(0.293 \times 10^5\,M^{-1}\,s^{-1})$, compared to the tetramer $(0.153 \times 10^5\,M^{-1}\,s^{-1})$, supports our proposed model in which the deubiquitinating activity in the USP25 tetramer assembly is partially blocked. It is important to note here the presence of dimer cross-contamination in the purified tetramer sample, as observed in a native gel (see Supplementary Figure 7b), which would increase the real activity of the tetramer sample. Interestingly, two USP25 point mutants that disrupt the tetramer assembly (F522G and E373A) display kinetic values close to the dimer of USP25 (Supplementary Figure 6d). Our results support a model in which the presence of dimer (active) or tetramer (inactive) regulates the deubiquitinating activity of USP25.

Strikingly, the absence of the whole LCC coiled-coil insertion in the monomer truncation, ΔLCC-IL USP25NCD, produced a notable reduction of the deubiquitinating activity, displaying a very low $K_{cat}/K_M$ $(0.055 \times 10^5\,M^{-1}\,s^{-1})$ (Fig. 5d). We attribute this catalytic impairment in the monomer to a wrong conformation of the catalytic domain, which probably requires contacts with the deleted LCC domain for a proper productive form of the protease (Supplementary Figure 9c). To assess this point, we generated a longer USP25 monomer including half of the LCC domain, named ΔLCC-IL USP25NCD long (Supplementary Figure 9), which displays a deubiquitinating activity similar to the USP25 dimer, thus fulfilling all contacts needed for a correct arrangement of the catalytic domain in the monomer.

**The USP25 dimer stabilizes tankyrases in cells**. We next checked the biological relevance of the tetramerization/inhibition mechanism of USP25 in cultured cells by checking the levels of endogenous tankyrases, which are bona fide USP25 substrates

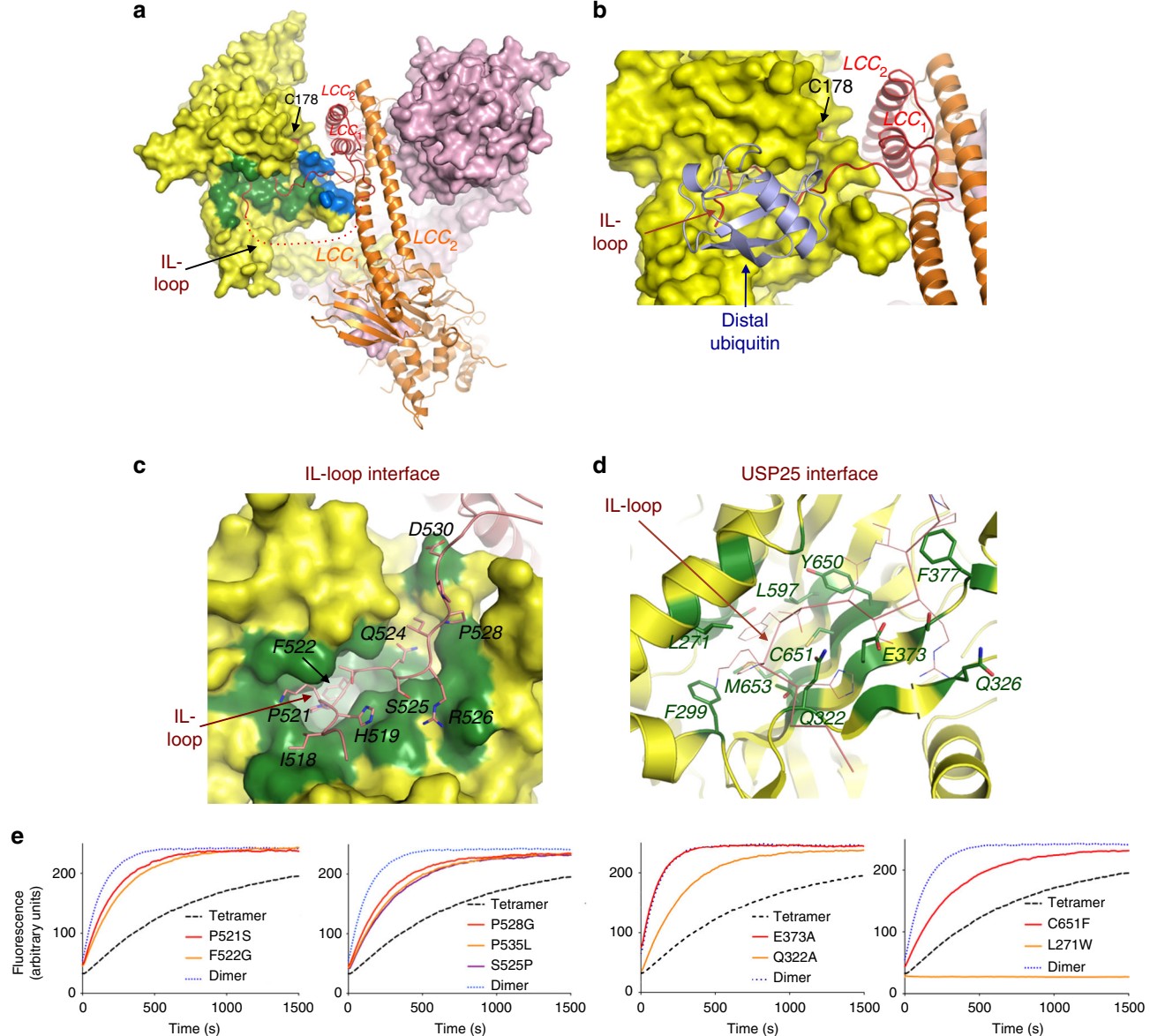

**Fig. 4** IL-loop contacts the catalytic domain of USP25 in the tetramer. **a** Surface and cartoon representation of interface between the IL-loop domain (green), "kink" region (blue), and the catalytic domain in the tetramer assembly. **b** Structure overlapping of USP25 and USP7 structures (PDB code 5JTJ), depicting the S1 ubiquitin-binding domain in contact with either ubiquitin in complex with USP7 structure (light blue) or with the IL-loop domain in the USP25 tetramer (red ribbon). **c** Surface representation of the S1 ubiquitin-binding pocket (green) of the USP25 catalytic domain in complex with the IL-loop insertion with labeled residues depicted in stick representation. **d** Cartoon representation of the IL-loop binding interface with labeled residues depicted in stick representation (green). IL-loop is depicted as a thin line (red). **e** Deubiquitinating activity assays with Ub-AMC substrate of either USP25 wild-type and IL-loop point mutants: purified tetramer and dimer, P521S, F522G, S525P, P528G, and P535L (first and second panel); or USP25 wild-type and S1 ubiquitin-binding residues point mutants: E373A, Q322A, L271W, and C651F (third and fourth panel)

involved in the regulation of the Wnt/β-catenin signaling[28]. Ectopic expression of wild-type and ΔIL truncation, either USP25FL or USP25NCD, in human embryonic kidney 293T (HEK293T) human cell lines revealed that the ΔIL USP25FL construct was able to increase significantly the protein levels of endogenous tankyrases after 48 h of expression, indicating a higher deubiquitinating activity of the constitutively dimer form of USP25 (Fig. 6a). Interestingly, the dimer assembly of USP25 lacking the CTD, ΔIL USP25NCD, which displayed similar in vitro deubiquitinating activities as USP25FL (Fig. 5c, d), did not increase the levels of takyrases, confirming the role of the CTD of USP25 in the specific interaction with tankyrases[28]. Also, an inactive USP25-active site mutant C178A produced similar

levels of tankyrases as the GFP control cultures, highlighting the role of the deubiquinating activity of the ectopically expressed USP25 in the tankyrase stability (Fig. 6a and Supplementary Figure 10a). These results were also observed by immunoprecipitation assays (Supplementary Figure 10b), which retrieved higher amounts of tankyrases by the dimer (ΔIL USP25FL) than by the tetramer (USP25FL). As expected, the C-terminal deletion constructs were unable to immunoprecipitate tankyrases (Supplementary Figure 10b). Also, cells treated with cyclohexamide, which inhibits novel protein synthesis, also revealed differences in the stability of tankyrases between dimer (ΔIL USP25) and tetramer (FL USP25) (Fig. 6b), probably due to a higher deubiquitinating activity of the "constitutive" dimer form of USP25.

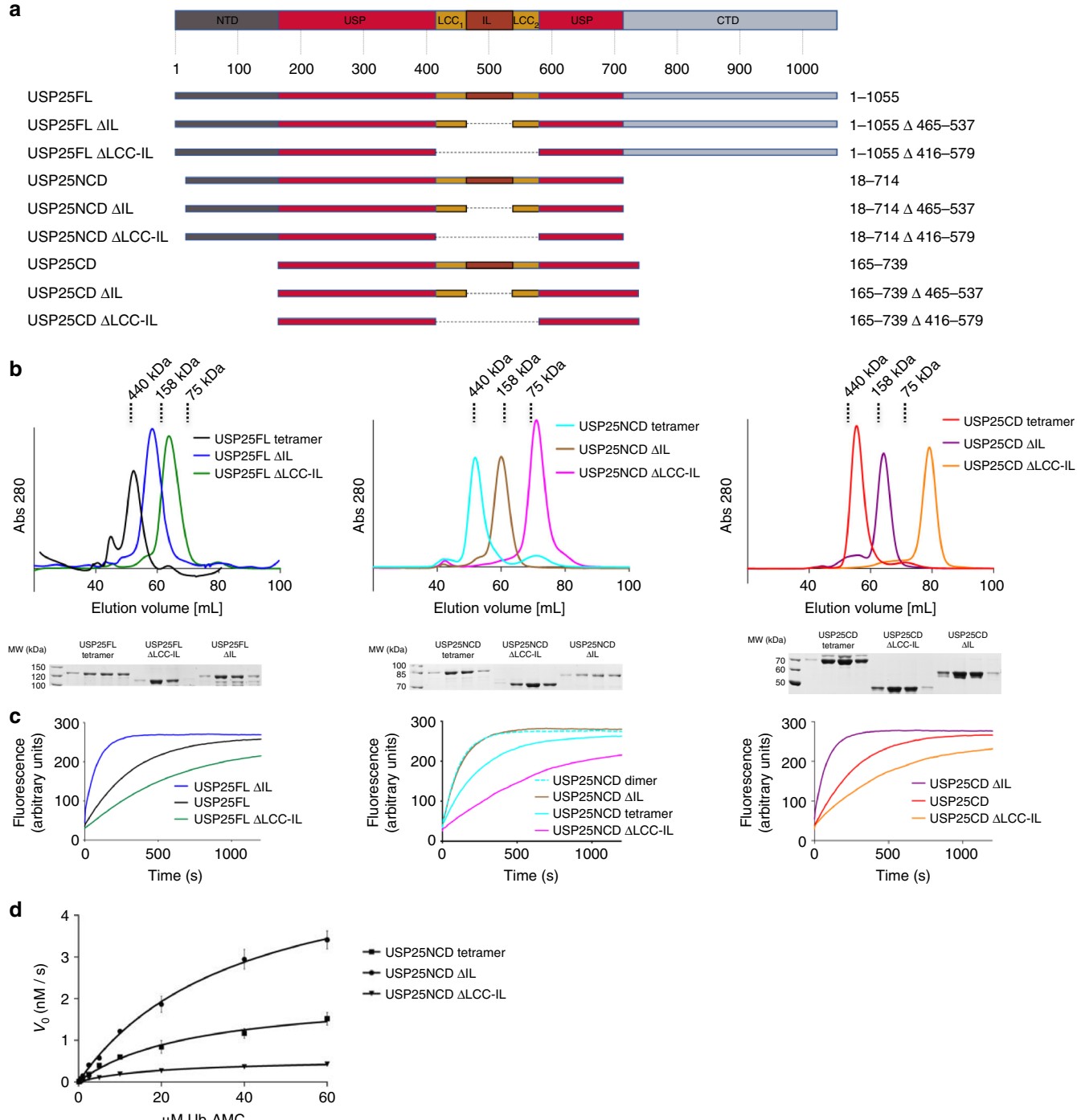

**Fig. 5** Biochemical and kinetic analysis of the USP25 assemblies. **a** Cartoon representation of the different truncation constructs of USP25. Left columns indicate the names and the right columns the residue range for each construct. **b** Size-exclusion chromatography profiles of the different USP25 truncation constructs run under identical experimental conditions in Superdex 16 column. Below, SDS-PAGE of the fractions of the gel filtration peaks. **c** Deubiquitinating activity assays of the different elution fractions of the gel filtration chromatography using Ub-AMC as a substrate. Activity assays were run in triplicate. **d** Steady-state "Michaelis–Menten" kinetic plot of the indicated oligomer constructs of USP25, including the purified tetramer, the truncated dimer construct (ΔIL) and the truncated monomer construct (ΔLCC-IL). Error bars represent the standard error for two independent experiments

### Table 2 Table of kinetic constant values for the USP25 oligomers

| USP25 constructs | $V_{max}$ (nM s⁻¹) | $K_M$ (μM) | $K_{cat}$ (s⁻¹) | $K_{cat}/K_M$ ($10^5$ M⁻¹ s⁻¹) |
|---|---|---|---|---|
| USP25NCD tetramer | 2.11 ± 0.16 | 27.72 ± 4.68 | 0.42 ± 0.03 | 0.153 |
| USP25NCD ΔIL | 5.66 ± 0.31 | 38.68 ± 4.20 | 1.13 ± 0.06 | 0.293 |
| USP25NCD ΔLCC-IL | 0.55 ± 0.03 | 20.17 ± 2.42 | 0.11 ± 0.01 | 0.055 |

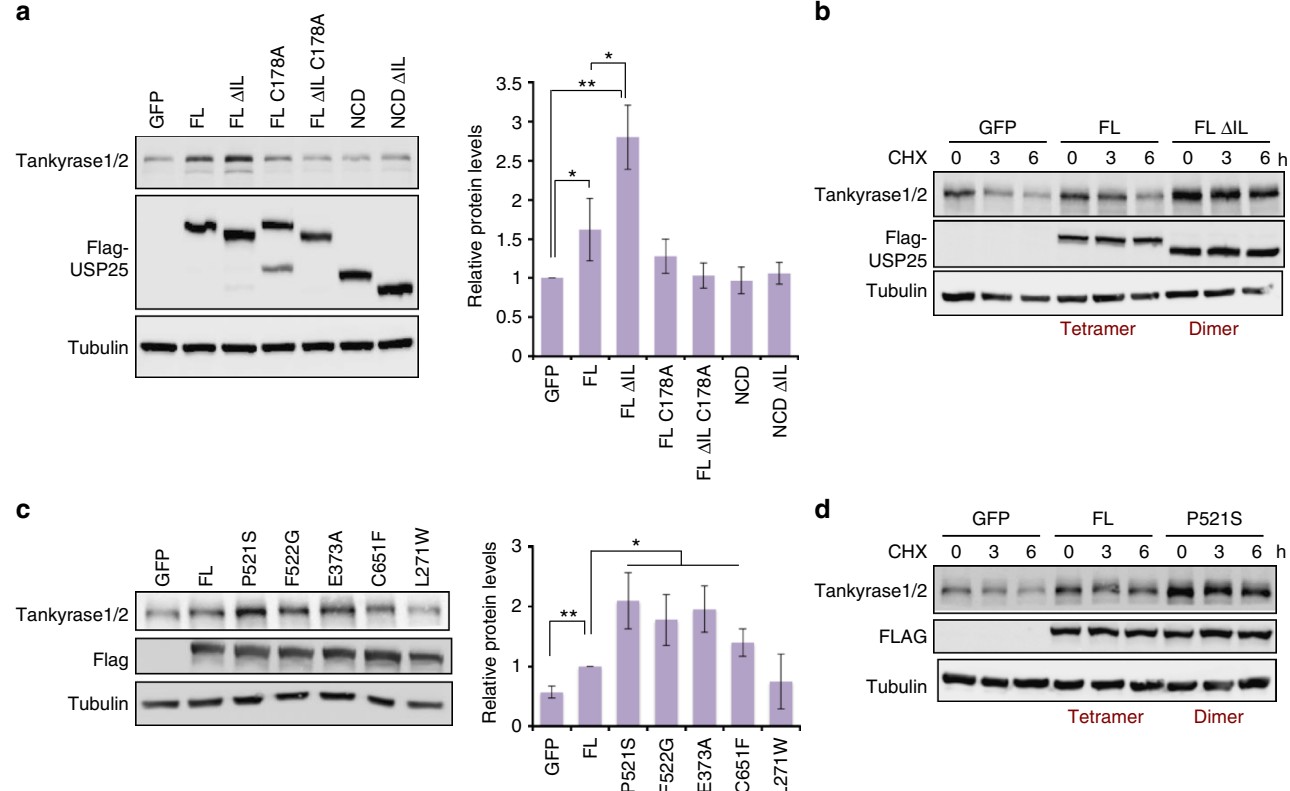

**Fig. 6** The USP25 dimer stabilizes endogenous tankyrase in HEK293T cells. **a** Flag-USP25FL, Flag-USP25FL ΔIL, Flag-USP25FL C178A, Flag-USP25FL ΔIL C178A, Flag-USP25NCD, and Flag-USP25NCD ΔIL were transfected in HEK293T cells and the levels of endogenous tankyrases1/2 were analyzed by western blot (WB). GFP was transfected as a control. Plot of the quantification of tankyrase1/2 levels, corrected with tubulin and relative to GFP. Data values are mean ± s.d. and $n \geq 3$ technical replicates. Significance was measured by a two-tailed unpaired $t$ test for all lanes relative to GFP and between USP25FL and USP25FL ΔIL. *$P < 0.05$, **$P < 0.01$. **b** HEK293T cells were transfected with Flag-USP25FL, Flag-USP25FL ΔIL, and GFP and cells were treated with 100 μg mL$^{-1}$ of cyclohexamide (CHX) and collected at indicated times for WB. Endogenous tankyrase1/2, Flag-USP25, and tubulin was checked and compared by WB. **c** Flag-USP25FL and indicated point mutants were transfected in HEK293T cells and the levels of endogenous tankyrases1/2 were analyzed by WB. Plot of the quantification of tankyrase1/2 levels relative to Flag-USP25FL. Data values are mean ± s.d. and at least $n \geq 3$ technical replicates. Significance was measured by a two-tailed unpaired $t$ test relative to FL. *$P < 0.05$, **$P < 0.01$. **d** HEK293T cells were transfected with Flag-USP25FL, Flag-USP25FL P521S, and GFP and cells were treated with 100 μgml$^{-1}$ of CHX and collected at indicated times for western blotting. Source data are provided as a Source Data file

Finally, ectopic expression of USP25 point mutants of the IL-loop interface, which weakens the tetramer assembly and impairs the deubiquitinating activity (Fig. 4e and Supplementary Figure 6), showed results comparable to the expression of the "constitutive" dimer at different levels (Fig. 6c). Only the L271W point mutant, which already showed a notable loss of activity in vitro, exhibited similar tankyrase levels as in the GFP control cultures. Cyclohexamide treatment of USP25 P521S point mutant yielded similar tankyrase stability as the USP25 dimer (Fig. 6d).

To confirm the ubiquitin-dependent stability of tankyrase[28], cells were treated with cyclohexamide and bortezomib, which is a potent proteasome inhibitor[35]. Bortezomib-treated cells revealed higher endogenous tankyrase levels in all cultures analyzed, including the GFP control and the poorly active USP25 L271W point mutant (Supplementary Figure 10c), all confirming the ubiquitin-dependent degradation of tankyrase[28] and the role of the deubiquitinating activity of USP25 for the higher stability of tankyrase.

All experiments support our in vitro characterization of the USP25 structure and might indicate a role of this tetramerization/inhibition mechanism of USP25 in a cellular context (Fig. 7).

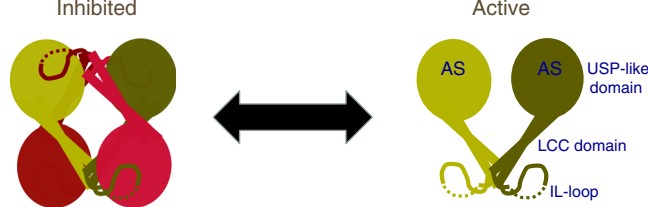

**Fig. 7** USP25 transition between dimer and tetramer. Each cartoon color represents a USP25 monomer. The scheme depicts a potential switch between tetramer and dimer assemblies of USP25, or between inhibited and active forms of USP25. AS active site

## Discussion

Two remarkable features are obtained from the crystal structure of USP25: the presence of two different stable quaternary assemblies, tetramer and dimer; and the implication of these two oligomer states in the deubiquitinating activity of USP25 by a unique autoinhibitory mechanism. This tetramerization-dependent inhibition mechanism of USP25 has never been described for other members of the DUB family and might

represent a paradigmatic example of regulation of the deubiquitinating activity, as we demonstrated in vitro and in cultured cells with the stabilization of tankyrases.

Among all described regulatory mechanisms for USPs, such as phosphorylation, allosteric binding of modulators, or localization, only two included comparable "autoregulation" mechanisms. For example: USP7 is activated by the interaction of a conserved C-terminal extension with the catalytic domain, producing a rearrangement of the catalytic active site triad;[11,12] and in USP4 the reaction product is released from the active site pocket thanks to the binding of an N-terminal ubiquitin-binding domain[13]. However, in USP25 the molecular mechanism is substantially different and the inhibition occurs by the contacts in the tetramer assembly.

USP25 is also regulated by post-translational modifications on its N-terminal domain, such as inhibition by SUMO conjugation[15]. The N-terminal domain of USP25, disordered in our crystal structure, contains ubiquitin-binding domains that are responsible for the interaction with ubiquitin chains, which can be probably perturbed by SUMO attachment. Our assays indicate that the N-terminal domain is required for an efficient deconjugation activity of ubiquitin chains, even in the case of the most "active" dimer (USP25CD ΔIL lane in Supplementary Figure 8), in contrast to the Ub-AMC substrate, in which the N-terminal domain is dispensable. Therefore, there are at least two different layers of USP25 regulation at the protein level: binding of ubiquitin chains to the N-terminal domain; and by a switch in the oligomerization state of USP25, described in the present work.

USP25, and its homolog USP28 (51% sequence identity) (Supplementary Figure 1b), are multidomain proteins characterized by the presence of long insertions in the middle of the USP-like catalytic domain, forming long coiled-coil structures (*LCC-IL-loop*) responsible for quaternary assemblies. This oligomerization domain is unique to USP25 (and USP28) in the USP family, in which no quaternary assemblies have been reported[30]. Moreover, PISA server analysis predicts the formation of the homotetramer structure ($A_4$) by the assembly of two distinctive homodimer structures ($2 \times A_2$), which we have validated by the characterization of several truncation constructs of USP25 (Fig. 6a, b). Interestingly, since the biophysical and enzymatic properties of the purified and truncated dimers are similar, we speculate whether a transition between these two oligomeric assemblies might be relevant in the regulation of the USP25 activity. Higher stability of endogenous tankyrases in HEK293T cells by the ectopic expression of a constitutive dimer (active) over a tetramer (inhibited) supports our regulatory tetramer/inhibition mechanism of USP25.

This regulatory mechanism depends on the tetramer assembly, with the interaction of the IL-loop insertion with the S1 ubiquitin-binding region of the catalytic domain. Remarkably, the IL-loop residues involved in the interaction are highly conserved across species, in contrast to the poor conservation displayed by the "non-observable" part of the IL-loop. In all our activity assays the tetramer assembly always displays a lower deubiquitinating activity in comparison to the dimer. In our work, almost all point mutants involved in the interface are assembled as dimers and displayed higher deubiquitinating activity. However, neither high ubiquitin concentrations cannot disrupt the tetramer assembly (Supplementary Figure 2d, e) nor increasing amounts the IL-loop peptide disturbs the USP25-deubiquitinating activity (Supplementary Figure 6e), excluding a direct competition between ubiquitin substrate and the IL-loop peptide. It is worth mentioning P535L point mutant, which does not directly contact the ubiquitin-binding pocket but its substitution by leucine probably affects the IL-loop orientation and thus decreases its binding affinity. P535L mutant was checked due its occurrence in three

independent cancer genomic studies (COSMIC database)[36]. Recently, USP25 has been associated to several types of cancer such as breast cancer, lung cancer, and non-small-cell lung cancer[25–27] and the P535L mutant would render USP25 more active than the wild type by destabilizing the tetramer assembly.

Our results reveal the presence of two stable assemblies of USP25, dimer and tetramer, or "active" and "inactive" enzyme, respectively. We believe that the lower activity observed in the tetramer assembly might be caused by dimer cross-contamination, as shown in native gels (Fig. 1 and Supplementary Figure S7). Our working model speculates on a post-translation modification that would promote a transition between tetramer and dimer in the cell (Fig. 6). In order to find relevant single-point mutations prompting to this oligomeric transition, we replaced USP25 Tyr454 either to phenylalanine or to glutamic acid. Tyr454 is located in the $LCC_1$ coiled-coil structure in contact with the "kink" motif and participates in the tetramer interface, being an excellent candidate to disassemble the tetramer (Supplementary Figure 11). Moreover, the conserved homolog residue in USP28 (Tyr447) has been reported to be phosphorylated in a human proteomics phospho-site global analysis[37]. Interestingly, in vitro analysis reveals that Y454F do not affect the tetramer stability, but Y454E, results in the formation of a stable dimer (Supplementary Figure 11). Thus, even though the presence of high number of contacts in the tetramer, a single point mutant in the interface, such as Y454E, is capable to destabilize the tetramer and render USP25 to an "active" dimer. Phosphorylation of a single residue of the USP catalytic domain has already been shown to regulate the deubiquitinating activity in USP14 and USP37[38,39], so we speculate whether it would be a plausible mechanism to switch between these two oligomer assemblies in USP25.

In addition to USP25, which is involved in the Wnt/β-catenin signaling by the regulation of the levels of tankyrases, the deubiquitinating activity of its homolog USP28 has been recently described to regulate the stability of P53[40–42]. Intriguingly, USP28 shares a high degree of sequence and structural homology with USP25 (Supplementary Figure 1b), so it would be reasonable that the inhibitory regulatory mechanism described here for USP25 might be extensive to USP28, thus adding another layer of complexity to the regulation of P53.

Finally, the inhibition of the USP25 activity by the interaction with the IL-loop, which basically comprises eight to ten residues sequence stretch, suggests the possibility of identifying small molecules that could mimic this specific binding to the USP25 catalytic domain. Those compounds would represent novel therapeutic approaches for the treatment of pathologies derived from an abnormal proliferative functions, such as the Wnt/β-catenin signaling pathway[43].

## Methods

**Plasmids and mutagenesis**. The full length of USP25 was cloned from pENTR-USP25 (purchased from Open Biosystems, Human ORFeome Collection). Different constructs of USP25 were amplified by PCR and cloned into the *Bam*HI/*Not*I restriction enzymes sites of pET28-Smt3 vector using ligation-dependent cloning, yielding an expression construct with an N-terminal SENP-cleavable Smt3 tag. Truncation mutants ΔIL and ΔLCC-IL were generated by overlapping extension PCR[45] and linked by a three-amino-acid linker peptide Gly-Ser-Gly. Point mutations were created using the QuickChange Site-Directed Mutagenesis Kit (Stratagene). All primer sequences are shown in Supplementary Table 1.

**Protein expression and purification**. pET28-Smt3 vectors harboring different constructs of USP25 were transformed into *Escherichia coli* strain Rosetta (DE3) (Novagen) cells and expression was induced with 0.5 mM isopropyl β-D-1-thio-galactopyranoside and grown at 28 °C overnight. Proteins were purified from the soluble cell lysate by $Ni^{2+}$ affinity chromatography using a buffer containing 20 mM Tris-HCl, pH 8.0, 250 mM NaCl, and 1 mM β-mercaptoethanol, and eluted with a similar buffer containing 250 mM imidazole. After Smt3 tag removal using

SENP2, the proteins were further purified by gel filtration (Superdex 200 column, GE Healthcare) chromatography pre-equilibrated in 100 mM NaCl, 20 mM Tris-HCl, pH 8.0, 1 mM β-mercaptoethanol, followed by an anion exchange (Resource Q column, GE Healthcare) chromatography, eluted with a linear gradient of NaCl (1 M NaCl, 20 mM Tris-HCl, pH 8.0, 1 mM β-mercaptoethanol). Proteins were concentrate using Amicon Ultra-30K ultrafiltration device (Milipore) prior to the following experiments.

**Methylation and crystallization.** USP25 construct containing residues 18–714 (USP25NCD) was expressed and purified by Ni$^{2+}$ affinity chromatography. After SENP2 cleavage, it was purified by gel filtration chromatography using Superdex 200 column pre-equilibrated in 250 mM NaCl, 50 mM HEPES, pH 7.5, 1 mM β-mercaptoethanol. Fractions containing the proteins of tetramer and dimer were pooled, followed by a lysine methylation step based on a published strategy[31]. In brief, borane–dimethylamine complex (Sigma-Aldrich) and for-maldehyde (Sigma-Aldrich) were sequentially added into protein solution and incubated overnight at 4 °C overnight. The methylation reaction was stopped by a final gel filtration chromatography on a Superdex 200 column pre-equilibrated in 200 mM NaCl, 20 mM Tris-HCl, pH 7.5, 1 mM β-mercaptoethanol, followed by an anion exchange chromatography. USP25NCD tetramer and USP25NCD dimer were finally concentrated to 8 g L$^{-1}$ for crystallization. Both USP25NCD tetramer and dimer could grow crystals, but only tetramer crystals could diffract well. Crystals were grown at 18 °C by hanging-drop vapor diffusion method by mixing the 1 μl protein in 100 mM NaCl, 10 mM Tris-HCl, pH 8.0, 1 mM β-mercaptoethanol with an equal volume of reservoir solution containing 18% PEG3350 (w v$^{-1}$), 100 mM Bis-Tris propane, pH 8.5, and 200 mM NaF. Full size crystals were obtained after 3 days, followed by a post-crystallization dehy-dration procedure, which involved transferring crystals to new solutions containing gradually increasing concentration of PEG3350 (18–25%). Diffraction-quality crystals were soaked in buffers supplemented with 15% (v v$^{-1}$) ethylene glycol and flash-frozen in liquid nitrogen. Crystals have unit cell dimensions of $a = 140.8$ Å, $b = 140.8$ Å, $c = 190.1$ Å, $\alpha = \beta = \gamma = 90°$. The attempts to find molecular replacement solutions using all previous published USP structures as search models did not generate any reliable solutions. Therefore, experimental phasing was performed by soaking the crystals in a drop containing 1 mM HgCl$_2$ and incubated overnight at 18 °C and then harvested from the soaking condition. Native and derivative datasets were collected at ALBA synchrotron in Barcelona (BL13-XALOC beamline)[46] and processed with XDS[47] and scaled, reduced, and further analyzed using CCP4[48].

**Structure determination and refinement.** SAD dataset was collected at 4.37 Å resolution using a wavelength of 0.9998 Å. Heavy atom sites and initial density map were calculated using PHENIX SAD Autosol[49]. The density maps clearly indicated the presence of three Hg$^{2+}$ sites bound to free cysteine residues, which allowed the formation of a rough initial density map. Density modification and phase extension programs from CCP4[48] were used to improve the initial density maps and build an initial model with Coot[50]. The initial model calculated with the Hg$^{2+}$ dataset was used as a search model for molecular replacement in the native dataset collected at 3.28 Å resolution with Phaser-Phenix[51]. The model was manually reconstructed using Coot[50] and further refine with PHENIX[49].

**Analytical gel filtration and native PAGE Assay.** All USP25 constructs were loaded on a Superdex 16 column pre-equilibrated in 100 mM NaCl, 20 mM Tris-HCl, pH 8.0, 1 mM β-mercaptoethanol and peak fractions were assayed by sodium dodecyl sulfate-polyacrylamide gel electrophoresis (SDS-PAGE). Curves were generated by plotting the absorbance at 280 nM vs. elution volumes and normal-ized to the same level. For running the native PAGE, all USP25 constructs were adjusted to 0.6 g L$^{-1}$ and prepared in non-reducing non-denaturing loading buffer. All gels were stained with Coomassie Brilliant Blue.

**Activity assays with diubiquitin and polyubiquitin chains.** Both diubiquitin (K48-linked) and polyubiquitin chain (K63-linked and K48-linked) hydrolysis reactions were performed in 150 mM NaCl, 20 mM Tris-HCl, pH 8.0, 5 mM dithiothreitol (DTT), 0.1% (v v$^{-1}$) Tween-20 with different USP25 constructs at 30 °C or 37 °C. The concentrations of enzymes and substrates in reactions were determined by specific experiments. Deubiquitylation of diubiquitin chains with native and lysine methylated USP25NCD tetramer was performed in 0.01 mg mL$^{-1}$ K48-linked diubiquitin substrates and 200 nM enzymes. Deubiquitylation of diu-biquitin chains with USP25NCD point mutants was performed in 0.025 mg mL$^{-1}$ diubiquitin (K48-linked) and 200 nM enzymes. Deubiquitylation of diubiquitin chains with USP25NCD truncation mutants was performed in 0.01 mg mL$^{-1}$ diubiquitin (K48-linked) and 100 nM enzymes. Deubiquitylation of polyubiquitin chains was performed in 0.05 mg mL$^{-1}$ K63-linked or K48-linked polyubiquitin chain substrates and 100 nM USP25 constructs. All reactions were stopped with SDS-loading buffer at the indicated times and analyzed by SDS-PAGE followed by staining with SYPRO (Bio-Rad).

**Dynamic light scattering.** The average size of all USP25 constructs were measured by DLS using a Malvern Zetasizer Nano-S90. Samples were measured in 100 μL

buffer with 150 mM NaCl, 20 mM Tris-HCl, pH 8.0, and 1 mM β-mercaptoethanol at 25 °C in 0.1 mL Malvern disposable polystyrene cuvettes. Analyses were performed in triplicates with high consistency and the representative results were shown. Due to aggregations, the intensity distributions that indicates how much light is scattered from various size "slices" or "bins" showed more than one peaks, and resulted in a high overall polydispersity index. However, when transforming the intensity distribution to a volume distribution, the result only showed a single peak. The volume contributions from the aggregation peaks were then so small that they were no longer displayed. The size distribution by volume graphs for different USP25 constructs were generated using GraphPad Prism 7.0.

**Hydrolysis and kinetic analysis with ubiquitin-AMC assay.** Ub-AMC hydrolysis assays and kinetic analysis were set up in fluorescence cuvettes (Hellma Analytics) in 100 μl buffer with 150 mM NaCl, 20 mM Tris-HCl, pH 8.0, 5 mM DTT, 0.1% (v v$^{-1}$) Tween-20 and measured with a fluorescence spectrometer (JASCO FP-8200) at excitation and emission wavelengths of 355 and 455 nM. Hydrolysis assays were performed at 37 °C with 200 nM USP25 constructs and 0.1 μM Ub-AMC in triplicates. Kinetic analysis (Michaelis–Menten kinetic measurements) were carried out using 5 nM USP25 constructs with a series of Ub-AMC substrate titrations at 37 °C. Initial rates of substrate hydrolysis were obtained using linear regression. Kinetic curves were obtained by plotting the measured enzyme initial rates vs. the corresponding substrate concentrations, followed by the modeling using nonlinear regression fit with Michaelis–Menten equation. All data were processed using GraphPad Prism 7.0.

**Cell culture analysis.** The HEK293T cell line (CRL-1573; ATCC) was used for ectopic expression of Flag-USP25 and its mutants. HEK293T cells were cultured in Dulbecco's modified Eagle's medium (Sigma-Aldrich, St Louis, MO, USA), sup-plemented with 10% (v v$^{-1}$) fetal bovine serum, 2μM L-glutamine, and 100 U mL$^{-1}$ penicillin/streptomycin. Cells were grown in a 37 °C humidified incubator con-taining 5% CO$_2$. The commercial antibodies used for western blotting (WB) analysis included the following: anti-TNKS1/2 (1:1000 dilutions; Santa Cruz Bio-technology; sc-8337); anti-USP25 (1:1000 dilution; Abcam, ab187156); anti-Flag (1:1000 dilutions; Sigma-Aldrich, F7425); and anti-Tubulin (1:5000 dilutions; Sigma, T5168), anti-Flag M2 affinity gel (Sigma-Aldrich, no. A2220). CHX (Sigma-Aldrich, no. 01810) was added to the cell culture medium in a final concentration of 100 μg mL$^{-1}$ and cells were collected at the indicated times (0, 3, and 6 h) for WB. Bortezomib (Jansen Pharmaceuticals) was added to the cell culture medium in a final concentration of 0.5 μM and cells were collected after 6 h for WB.

**Cell culture vector constructs and transfection.** pcDNA3.1-Flag-USP25FL, pcDNA3.1-Flag-USP25FL ΔIL, pcDNA3.1-Flag-USP25FL C178A, pcDNA3.1-Flag-USP25FL ΔIL C178A, pcDNA3.1-Flag-USP25NCD, pcDNA3.1-Flag-USP25NCD ΔIL, and pcDNA3.1-Flag-USP25FL point mutants P521S, F522G, E373A, C651F, and L271W vector constructs were generated by PCR using previous pET28-Smt3 USP25 constructs as DNA templates, the specific primers fused with a Flag tag at the N Termini and Phusion DNA polymerase (Thermo) following the manu-facturer's instructions. The PCR products were cloned into pcDNA3.1 (Invitrogen) using the T4DNA ligase (Thermo) and sequenced previous to use. pcDNA3.1-Flag-USP25 and its mutated vector constructs were transfected into HEK293T cells using the Lipotransfectin (NIVORLAB), according to the manufacturer's instruc-tions. Forty-eight hours later, transfected HEK293T cells were collected and lysed in Triton lysis buffer (TLB: 50 mM Tris-HCl, pH 7.5, 150 μM NaCl, 1 μM EDTA, 50 μM NaF, 0.5% Triton X-100, plus protease inhibitors).

**WB experiments.** Total protein lysates derived from transfected HEK293T cells were used for WB experiments using the human specific antibodies anti-TNKS1/2 (1:1000 dilutions; Santa Cruz Biotechnology; sc-8337); anti-USP25 (1:1000 dilu-tion; Abcam, ab187156); anti-Flag (1:1000 dilutions; Sigma-Aldrich, F7425); and anti-Tubulin (1:5000 dilutions; Sigma, T5168). Membranes were developed with chemiluminescence substrate Pierce® ELC Western Blotting Substrate (Thermo Fisher Scientific, South Logan, UT, USA), and visualized on a LAS4000 device (Fujifilm, Tokyo, Japan). Protein quantification was done with the Image Gauge software (Fujifilm).

**Immunoprecipitation experiments.** Total protein extracts obtained from HEK293T cells were transiently transfected with Flag-USP25 or its mutant expression plasmids using TLB as describer before. Five milligram of protein extracts were used for immunoprecipitation (IP) using specific anti-Flag M2 affinity gel. IP was performed overnight at 4 °C with slow rotation. After 24 h, the supernatant was separated from the gel and washed three times with TLB, the Flag-fused proteins were eluted with SDS-loading buffer for 1 h at room temperature. Immunoprecipitated proteins were analyzed by WB using the specific anti-Flag antibodies and anti-TNKS1/2.

**Statistical analysis for table of tankyrase stability.** Data are represented as mean ± SD of three or more independent experiments. Statistical tests were per-formed using LAS4000 and ImageJ 1.49v softwares. Comparison between samples

was evaluated by independent sample *t* test, and results were considered statistically significant when *p* value <0.05

## Data availability

Coordinates and reflection data files have been deposited in the Protein Data Bank with accession number 5O71. A reporting summary for this article is available as a Supplementary Information file. The source data underlying Figs. 6a and c are provided as a Source Data File. Other data are available from the corresponding author upon reasonable request.

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

## Acknowledgements

This work was supported by grants from the "Ministerio de Economia y Competitividad" BFU2015-66417-P (MINECO/FEDER) to D.R. and BFU2015-64879-R to V.A., B.L. and Y.Z. acknowledge the support from the China Scholarship Council (China).

## Author contributions

B.L. conducted all protein expression, crystallization, and kinetics experiments. Y.Z. conducted the initial protein expression and crystallization. B.L. and D.R. solved the crystal structure. V.A. and M.S.-G. conducted in vivo cell culture analysis. D.R. and B.L. analyzed the results and wrote the paper. D.R. conceived the idea for the project.

## Additional information

**Competing interests:** The authors declare no competing interests.

