## [Peer Review File · Nature Communications]

Reviewers' comments:

Reviewer #1 (Remarks to the Author):

In this manuscript Liu et al. reported the structure of USP25 and biochemical characterizations of its enzyme activity. The formation of a tetrameric structure was observed and attributed to the difference in the observed enzyme activities of the different forms of USP25. This is an interesting study and demonstrates a potential autoinhibition of DUB enzyme activity. Given that crystallography can lead to oligomeric structures due to crystallographic packing, a clear demonstration of the functional relevance of this observation is key. The authors have carried out studies to assess this scenario. However, further studies are necessary to lend important support of the proposed mechanism and provide a better understanding of the system.

1. The lysine methylation of USP25 was done to improve the resolution of the protein. This is a nonspecific chemical modification of many lysine residues in USP25. Did the author test the methylated USP25 to ensure that the USP25 functions were not affected? Given that methylation changes the side-chain size, hydrogen bonding and charge interactions, can the author be sure that it did not affect the protein structures and the cross-protomer interactions?
2. Figure 3b can be colored differently to distinguish the residues from the two different protomers. Currently it is hard to read.
3. A complete kinetics analysis of the mutant F522G and P535L USP25 is needed to compare k_{cat} and K_m with that of the wild type USP25.
4. A reciprocal mutation analysis of the residues in the USP25 active site that are proposed to interact with the IL loop residues will provide good confirmation of the function of the proposed interactions in the tetramer formation.
5. In Fig. 5B, the MW of the various USP25 species needs to be calculated to assess the size of the proteins. Given that the overall shape of the protein species would change drastically due to the presumed different oligomeric structures, can the author corroborate the results with other techniques such as dynamic light scattering and SAXS.
6. Given that Ub-AMC is an artificial substrate and lacks other crucial interactions in deubiquitination. The authors should carry out kinetic analysis of the wild-type and mutant USP25 using polyubiquitin or diubiquitin substrates and determine whether the overall trend holds.
7. The observed difference between USP25NCD and USP25NCD Δ IL is fairly small. The deletion mutant USP25NCD Δ LCC-IL showed much reduced activity. This is hard to explain given the proposed mechanism of autoinhibition. The provided explanation was the possible change in the active site arrangement. The author should provide support of this claim.
8. In the cellular experiment, the endogenous USP25 was not knocked out based on the method description. This complicates the study because a mix of wild-type and mutant USP25 coexists in cells. For clarity the author should knock out (or knock down) endogenous USP25 and access the cellular activity and function of USP25.

Reviewer #2 (Remarks to the Author):

The manuscript from Liu et. al. "A quaternary tetramer assembly inhibits the deubiquitinating activity of USP25" analyses USP25 by biochemical and structural methods. Their analyses reveal that USP25 exists as a dimer and tetramer, and the tetrameric form shows reduced catalytic activity. Analysing the crystal structure of USP25, they find further evidence for tetramer formation and the molecular details of oligomerization. In the tetramer, a region in the unique insert within the catalytic domain binds to the ubiquitin binding site of the catalytic domain thereby inhibiting the enzyme. Based on these findings, the authors conclude that the dimer has higher activity compared to the tetramer. By sequence analysis, they propose that a tyrosine residue at the interface may be phosphorylated. Substitution of this tyrosine to a negatively charged amino

acid to mimic phosphorylation results in a dimer that is more active. In summary, they conclude that USP25 is inhibited by tetramer formation.

Overall, while this is an interesting finding and may present a new model for how activity of USP25 is regulated, it is still preliminary and requires extensive analyses to support their conclusions and provide better mechanistic insights (see below for specific comments and suggestions to improve the manuscript). Another major issue is that the manuscript suffers from poor organisation as a result of which there is no logical flow and it is all very jumbled and confusing to read.

Specific comments:

1. To make the manuscript more logical, it would help if the authors first describe the monomer of USP25, followed by how dimerization and tetramerization occur. Reorganization of figures and text along with better quality figures and cartoons to illustrate these findings along with close up views of tetramer interfaces will help.
2. The authors have not analysed where the proximal ubiquitin will sit. They can make a fair estimation of this by comparing structures of other USP DUBs in complex with diubiquitin.
3. Several mechanistic aspects of ubiquitin binding are unclear:
 - Since the IL loop interacts with the S1 ubiquitin binding pocket, how does ubiquitin displace this loop. In other words, what is the affinity of the loop versus that of ubiquitin for binding to the S1 pocket
 - Further, it will be important to evaluate how ubiquitin binding affects or influences oligomerization
4. The authors must discuss how they envision USP25 is activated, i.e. how it transitions from a tetramer to a dimer to be active. I appreciate that any experiments to address this may be beyond the scope of this present manuscript but any additional effort in this direction will vastly strengthen this present study.
5. All enzyme kinetics are performed with Ub-AMC. Using a more relevant substrate such as K63 diubiquitin will be more informative.
6. Rather than make large deletions right in the middle of the catalytic domain (Figure 5), the authors should use the structural insights to test minimal point mutants that disrupt tetramer assembly. What is the oligomerization status of the point mutants F522G and P535L that show increased activity?

Other comments:

- Many spelling mistakes and grammatical errors throughout the manuscript that made it very difficult to read.
- "Ubiquitination activates USP25 whereas SUMOylation inhibits USP25 activity". This statement is made in the introduction. As the focus of the manuscript is understanding how USP25 is regulated, it will be important to discuss these published findings in context of the inhibition by tetramer formation described here.
- RMSD of superposition in Figure S2 not mentioned. The close up view of catalytic centre could be moved to main figures. To make space, the stereo images which aren't very good can be moved to supplementary figures.
- Figure S5 is wrongly labelled as Figure S4
- Figure 2 labelling is incorrect in many parts. Please use different colours to distinguish NTD and IL in Figure 2A. RMSD values for USP25 and USP7 superposition
- To the non-specialist it is not clear how the authors arrived at where the S1 Ubiquitin binds. Figure 4A and 4D should sit next to each other, maybe with an intermediate figure in between which will then make it clear to the reader that the IL loop sits where the S1 Ub binds.
- Cyclohexamide is spelt everywhere as 'ciclohexamide'
- Crucial control of catalytically dead Cys-Ala mutant is missing in experiments shown in Figure 6. This is because the cells express endogenous wildtype USP25.
- As Tankyrase is a well-established substrate of USP25, it would be easy to purify ubiquitinated tankyrase (following proteasome inhibition) and use it as a substrate for assaying enzyme activity of USP25.

Reviewer #1:

1. The lysine methylation of USP25 was done to improve the resolution of the protein. This is a nonspecific chemical modification of many lysine residues in USP25. Did the author test the methylated USP25 to ensure that the USP25 functions were not affected? Given that methylation changes the side-chain size, hydrogen bonding and charge interactions, can the author be sure that it did not affect the protein structures and the cross-protomer interactions?

Response: The quality of USP25 after the methylation protocol has been checked and it is now displayed in the new **Supplementary figure S2a**. In this figure we have compared the gel filtration profiles of the USP25 tetramer before and after the methylation protocol. In both cases the elution of USP25 in the gel filtration indicates that the quaternary assembly is maintained as a tetramer, as well as in a native PAGE analysis under non-denaturing conditions. The activity of the methylated USP25 was also checked against diUbiquitin K48-linked substrate (new **Supplementary figure S2b**), displaying a similar deubiquitinating activity as for the WT tetramer.

A new sentence has been added in page 6: “Apparently, the lysine methylation reaction did not affect substantially the integrity of USP25 (**Supplementary Fig. 2a**).”.

2. Figure 3b can be colored differently to distinguish the residues from the two different protomers. Currently it is hard to read.

Response: After the suggestions of the second reviewer, we have prepared a new set of figures to better display the different oligomer structures of USP25: please check new **Figures 2, 3 and 4**.

Old stereo **Figure 3b** has been moved to new **Supplementary figure S4a** and as required by the reviewer the residues numbers have been colored according to the protomer.

3. A complete kinetics analysis of the mutant F522G and P535L USP25 is needed to compare k_{cat} and K_m with that of the wild type USP25.

Response: Initial F522G and P535L point mutants were prepared as full length USP25 constructs (USP25 FL), but the complete kinetic analysis of the wild type USP25 was performed with a construct lacking the C-terminal domain (USP25 NCD). However, after suggestions by both reviewers, in addition to F522G and P535L point mutants (USP25 NCD constructs), we have prepared a whole set of new point mutants of the interface between the IL-loop and the USP-like binding domain. All these point mutants (including F522G and P535L) were newly prepared without the C-terminal domain (USP25 NCD) and in all cases the oligomerization state has been checked as well as the de-ubiquitinating activity (see next point 4). The complete kinetic analyses have been only performed with the USP25 truncation mutants, not with the 10 new point mutants of the interface, including a novel construct of the monomer in the revised version (see point 7).

4. A reciprocal mutation analysis of the residues in the USP25 active site that are proposed to interact with the IL loop residues will provide good confirmation of the function

of the proposed interactions in the tetramer formation.

Response: After the suggestions by both reviewers, we have prepared a new set of point mutants, including reciprocal mutation from the two surfaces, to check the role of IL-loop interaction with the USP-like catalytic domain. These mutants include: L271W, C651F, Q322A, E373A, Q322A/E373A, P521S, F522G, S525P, P528G, and P535L. The results are shown in new **Figure 4e**, which compares the activities towards the Ub-AMC substrate, and in new **Supplementary figure S6**, which displays the gel filtration analysis and the de-ubiquitinating activity against diUbiquitin K48-linked substrate.

A new paragraph has been added in page 9: “To assess the role of this interface, several USP25 point mutants were generated in the IL-loop and in the S1 ubiquitin-binding surface (**Fig. 4e**, and **Supplemental Fig. 6**). P521S and F522G IL-loop point mutants, which are involved in hydrophobic contacts within the pocket, disrupt the tetramer assembly and their deubiquitinating activity is comparable to the USP25 dimer. Similar results are observed for P528G, P535L and S525P IL-loop point mutants, but in this instance the distorted conformation of the IL-loop probably perturbs the binding affinity. On the USP-like catalytic domain surface, both E373A and Q322A point mutants of the S1 ubiquitin-pocket disrupt three hydrogen bonds with the IL-loop, which compromise the tetramer assembly and displays higher deubiquitinating activities. Finally, C651F and L271W point mutants were intended to close the USP-like binding cleft, preventing the IL-loop binding. Whereas this was probably achieved for C651F, L271W point mutant was purified as a tetramer with a completely abolished deubiquitinating activity. Overall, this reciprocal point mutant analysis verifies the role of the IL-loop in the tetramer assembly and in the deubiquitinating activity”.

5. In Fig. 5B, the MW of the various USP25 species needs to be calculated to assess the size of the proteins. Given that the overall shape of the protein species would change drastically due to the presumed different oligomeric structures, can the author corroborate the results with other techniques such as dynamic light scattering and SAXS.

Response: To confirm the MW of the different USP25 truncation constructs, in addition to the native SDS gels under non-denaturing conditions and to the analytical gel filtration analysis, we have conducted a *Dynamic Light Scattering (DLS)* analysis for all samples, as suggested by the reviewer, and the results are displayed in new **Supplementary figure S7d**. As observed in the DLS table, the estimated MW by DLS fits very well with the theoretical MW for all USP25 truncation constructs.

Alternatively, to make more comprehensive the analytical gel filtration figure, we have added in new **figure 5b** the elution times for standard protein markers.

A new sentence has been added in page 10: “Additionally, dynamic light scattering analysis of the USP25 truncation constructs also displayed a good correlation between oligomers and their theoretical molecular weight (**Supplementary Fig. 7**).

6. Given that Ub-AMC is an artificial substrate and lacks other crucial interactions in deubiquitination. The authors should carry out kinetic analysis of the wild-type and mutant

USP25 using polyubiquitin or diubiquitin substrates and determine whether the overall trend holds.

Response: As suggested by both reviewers, in addition to the Ub-AMC substrate, we have conducted a comparative de-ubiquitinating activity analysis for the USP25 truncation constructs with either polyUb K48- and K63-linked polyubiquitinated substrates, and with di-ubiquitin K48-linked substrate. All these results are displayed in the new **Supplementary figure S8**. The results confirm the higher activity for the USP25 dimer, as well as the role of the N-terminal domain in the binding of ubiquitin chains. K48-linked di-ubiquitin substrate has also been used to check the point mutants of the IL-loop binding to the USP-like domain (new **Supplementary figure S6d**).

A new sentence has been added in page 10: “*The higher activity of the USP25 dimer over tetramer was also confirmed using poly-ubiquitin chain, either K48 or K63-linked, and di-ubiquitin K48-linked substrates (Supplementary Fig. 8)*”.

7. *The observed difference between USP25NCD and USP25NCD Δ IL is fairly small. The deletion mutant USP25NCD Δ LCC-IL showed much reduced activity. This is hard to explain given the proposed mechanism of autoinhibition. The provided explanation was the possible change in the active site arrangement. The author should provide support of this claim.*

Response: Since we don't have the structure of the truncated monomer to check the catalytic triad arrangement, it is difficult to know why the catalysis is reduced. However, we have designed a longer monomer construct including the first half of the LCC domain, named Δ LCC-IL USP25NCD long, expecting that the all contacts between the LCC domain and the USP-like domain will be maintained and thus the proper arrangement of the active site. These contacts were disrupted in the first designed monomer construct (Δ LCC-IL). The kinetic results in the new **figure 5d** revealed a K_{cat} constant similar to the dimer and tetramer, indicating that the new longer monomer construct is as catalytically competent as the wild type. This new construct, Δ LCC-IL USP25NCD long, has been also characterized in the new **Supplementary Fig. 9** by gel filtration and DLS analysis.

A new paragraph has been added in page 10: “*Strikingly, the absence of the whole LCC coiled-coil insertion in the monomer truncation, Δ LCC-IL USP25NCD, produced a notable reduction of the catalytic constant ($K_{cat} = 0.07 \pm 0.02 \text{ s}^{-1}$) but did not affect the K_M constant for ubiquitin-AMC (Fig. 5d). We attribute this catalytic impairment to a wrong arrangement of the active site in the monomer, which probably requires interacting with the deleted LCC domain. To assess this point, we generated a longer USP25 monomer including half of the LCC domain, named Δ LCC-IL USP25NCD long (Supplementary Fig. 9), which displays a catalytic constant ($K_{cat} = 0.42 \pm 0.07 \text{ s}^{-1}$) similar to the USP25 dimer and tetramer (Fig. 5d), thus fulfilling all contacts needed for the proper arrangement of the catalytic domain*”.

8. *In the cellular experiment, the endogenous USP25 was not knocked out based on the method description. This complicates the study because a mix of wild-type and mutant*

USP25 coexists in cells. For clarity the author should knock out (□or knock down)□ endogenous USP25 and access the cellular activity and function of USP25.

Response: The role of the endogenous USP25 in HEK293 cells has been previously analyzed in “*Xu et al., USP25 regulates Wnt signaling by controlling the stability of tankyreses. Genes and Dev. 31: 1-12 (2017).*” Alternatively, we have checked the levels of endogenous USP25 in HEK293 cells in a new WB blot with anti-USP25 Ab (new **Figure 6a**). We observed that the levels of endogenous USP25 are fairly small compared to the ectopic USP25 and are equally distributed in all cultures, including the negative culture control (GFP). Our results, together with the previously published in *Genes and Dev. 31: 1-12 (2017)*, indicate that the effect of the endogenous USP25 in the tankyrase stability, compared to the ectopically expressed USP25, is residual.

Alternatively, we have prepared two new constructs with an active site cysteine mutant, C178A, which completely abolishes the catalytic activity of the ectopically expressed USP25 (new **Fig. 6b**). Interestingly, the C178A mutant of the constitutive USP25 dimer does not stabilize tankyrase as the wild type.

A new sentence has been added in page 11: “*Also, the inactive USP25 active site mutant C178A produced similar levels of tankyrases as the control cultures, highlighting the role of the ectopic expression of USP25 in the tankyrase stability (Fig. 6b)*”.

Reviewer #2:

Specific comments:

1. To make the manuscript more logical, it would help if the authors first describe the monomer of USP25, followed by how dimerization and tetramerization occur. Reorganization of figures and text along with better quality figures and cartoons to illustrate these findings along with close up views of tetramer interfaces will help.

Response: Sorry for that. We have substantially modified the order of the manuscript to make it more comprehensible to the reader, as suggested by the reviewer. We have also prepared a new set of figures, which we hope they illustrate better all points described in the text (including close up views of the interfaces). Please check new **Figures 2,3,4** and new **Supplementary Fig. 3, 4**.

2. The authors have not analysed where the proximal ubiquitin will sit. They can make a fair estimation of this by comparing structures of other USP DUBs in complex with diubiquitin.

Response: We have analyzed this point and it is now displayed in new **Supplementary Fig. 3b**. USP25 has been overlapped with CYLD-K63 diUbiquitin structure (rmsd 2,74 Å over 218 residues) (PDB code 3WXG), and the “proximal” and “distal” ubiquitins have been modeled on the USP25 dimer, displaying few steric clashes.

3. Several mechanistic aspects of ubiquitin binding are unclear:

- Since the IL loop interacts with the S1 ubiquitin binding pocket, how does ubiquitin displace this loop? In other words, what is the affinity of the loop versus that of ubiquitin for binding to the S1 pocket?

Response: We have analyzed this point in the new **Supplementary Fig. 2b**, by checking the competition of free ubiquitin and the IL-loop for the S1 ubiquitin-binding surface in the tetramer assembly. Neither the USP25 tetramer assembly, nor the de-ubiquitinating activity is substantially disturbed by the presence of high amounts of ubiquitin. These results indicate a stronger affinity of the IL-loop for this surface than ubiquitin.

A new sentence has been added in page 13: *“Interestingly, the binding affinity of IL-loop for this surface seems higher than for ubiquitin, as shown in a competition analysis (Supplementary Fig. 2b) where high ubiquitin concentrations can not disrupt the tetramer assembly”*.

- Further, it will be important to evaluate how ubiquitin binding affects or influences oligomerization.

Response: As displayed in new **Supplementary Fig. 2b**, the USP25 tetramer assembly is not affected by the presence of a high ubiquitin concentration. The activation seems to rely on other factors that can destabilize the tetramer assembly. This point is discussed at the end of the discussion and in next point 4, in which phosphorylation of an interface residue might trigger the activation by promoting a oligomer transition.

4. The authors must discuss how they envision USP25 is activated, i.e. how it transitions from a tetramer to a dimer to be active. I appreciate that any experiments to address this may be beyond the scope of this present manuscript but any additional effort in this direction will vastly strengthen this present study.

Response: As we discussed at the end of the manuscript, a single change of a residue charge involved in the tetramer interface, such as phosphorylation, might suffice for the tetramer disassembly and activation. This is a very intriguing point and a major goal of research in my group: to figure out the switch to assemble or disassemble the USP25 tetramer.

5. All enzyme kinetics are performed with Ub-AMC. Using a more relevant substrate such as K63 diubiquitin will be more informative.

Response: As suggested by both reviewers, in addition to the Ub-AMC substrate, we have conducted a comparative de-ubiquitinating activity analysis for the USP25 truncation constructs with either polyUb K48- and K63-linked polyubiquitinated substrates, and with di-ubiquitin K48-linked substrate. All these results are displayed in the new **Supplementary figure S8**. The results confirm the higher activity for the USP25 dimer, as well as the role of the N-terminal domain in the binding of ubiquitin chains. K48-linked di-ubiquitin substrate has also been used to check the point mutants of the IL-loop binding to the USP-like domain (new **Supplementary figure S6d**).

A new sentence has been added in page 10: *“The higher activity of the USP25*

dimer over tetramer was also confirmed using poly-ubiquitin chain, either K48 or K63-linked, and di-ubiquitin K48-linked substrates (Supplementary Fig. 8)”.

6. Rather than make large deletions right in the middle of the catalytic domain ((Figure 5)), the authors should use the structural insights to test minimal point mutants that disrupt tetramer assembly. What is the oligomerization status of the point mutants F522G and P535L that show increased activity?

Response: After the suggestions by both reviewers, we have prepared a new set of point mutants, including reciprocal mutation from the two surfaces, to check the role of IL-loop interaction with the USP-like catalytic domain. These mutants include: L271W, C651F, Q322A, E373A, Q322A/E373A, P521S, F522G, S525P, P528G, and P535L. The results are shown in new **Figure 4e**, which compares the activities towards the Ub-AMC substrate, and in new **Supplementary figure S6**, which displays the gel filtration analysis and the de-ubiquitinating activity against diUbiquitin K48-linked substrate.

A new paragraph has been added in page 9: *“To assess the role of this interface, several USP25 point mutants were generated in the IL-loop and in the S1 ubiquitin-binding surface (Fig. 4e, and Supplemental Fig. 6). P521S and F522G IL-loop point mutants, which are involved in hydrophobic contacts within the pocket, disrupt the tetramer assembly and their deubiquitinating activity is comparable to the USP25 dimer. Similar results are observed for P528G, P535L and S525P IL-loop point mutants, but in this instance the distorted conformation of the IL-loop probably perturbs the binding affinity. On the USP-like catalytic domain surface, both E373A and Q322A point mutants of the S1 ubiquitin-pocket disrupt three hydrogen bonds with the IL-loop, which compromise the tetramer assembly and displays higher deubiquitinating activities. Finally, C651F and L271W point mutants were intended to close the USP-like binding cleft, preventing the IL-loop binding. Whereas this was probably achieved for C651F, L271W point mutant was purified as a tetramer with a completely abolished deubiquitinating activity. Overall, this reciprocal point mutant analysis verifies the role of the IL-loop in the tetramer assembly and in the deubiquitinating activity”.*

Other comments:

- Many spelling mistakes and grammatical errors throughout the manuscript that made it very difficult to read.

Response: We apologize for the lack of accuracy with some expressions. We hope that it has been corrected in the new version.

- "Ubiquitination activates USP25 whereas SUMOylation inhibits USP25 activity". This statement is made in the introduction. As the focus of the manuscript is understanding how USP25 is regulated, it will be important to discuss these published findings in context of the inhibition by tetramer formation described here.

Response: We have added in the Discussion section a new paragraph about the USP25 regulation by posttranslational modification (SUMO) and by our tetramerization/inhibition

mechanism. Basically, we believe that SUMO or ubiquitin modification at the N-terminal domain regulates the binding to ubiquitin chains to the N-terminal domain, as shown in new **Supplemental Fig. 8**. However, the switch between tetramer and dimer, revealed in this work, is an unrelated mechanism to activate the protease.

A new paragraph has been added in page 12: *“USP25 is also regulated by posttranslational modifications on its N-terminal domain, such as inhibition by SUMO conjugation. The N-terminal domain of USP25, disordered in our crystal structure, contains ubiquitin-binding domains that are responsible for the interaction with ubiquitin chains, which can be probably perturbed by SUMO attachment. Our assays indicate that the N-terminal domain is required for an efficient de-conjugation activity of ubiquitin chains, even in the case of the most “active” dimer (USP25CD Δ IL lane in **Supplementary Fig. 8**), in contrast to the Ub-AMC substrate, in which the N-terminal domain is dispensable. Therefore, there are at least two independent layers of USP25 regulation at the protein level: binding of ubiquitin chains to the N-terminal domain; and by a switch in the oligomerization state of USP25, described in the present work”.*

- RMSD of superposition in Figure S2 not mentioned. The close up view of catalytic centre could be moved to main figures. TO make space, the stereo images which aren't very good can be moved to supplementary figures.

Response: RMSD values have been added in the main text. The close up view has been moved to the main figure (new **Figure 2d**). Stereo images have been all moved to the new **Supplementary Fig. S4**.

- Figure S5 is wrongly labelled as Figure S4

Response: Sorry for that. It has been corrected.

- Figure 2 labelling is incorrect in many parts. Please use different colors to distinguish NTD and IL in Figure 2A. RMSD values for USP25 and USP7 superposition

Response: Sorry for that. We have double-checked the figure. We have modified the colors of panel A in new **Figure 2**, as suggested by the reviewer. RMSD values have been added in the main text.

A new sentence has been added in page 7: *“which is comparable to the USP7-Ub aldehyde complex structure (rmsd 1,67 Å for 268 aligned residues)”.*

- To the non-specialist it is not clear how the authors arrived at where the S1 Ubiquitin binds. Figure 4A and 4D should sit next to each other, maybe with an intermediate figure in between which will then make it clear to the reader that the IL loop sits where the S1 Ub binds.

Response: As suggested by the reviewer, we have placed panels A and B next to each other in new **Figure 4**. We hope now it will be clearer to the reader the competition of IL-loop and ubiquitin for the same S1 surface in the USP-like domain.

- Cyclohexamide is spelt everywhere as 'ciclohexamide'

Response: Sorry for that. It has been corrected.

- Crucial control of catalytically dead Cys-Ala mutant is missing in experiments shown in Figure 6. This is because the cells express endogenous wildtype USP25.

Response: As suggested by the reviewer, we have prepared two new constructs including an inactive point mutant of the cysteine of USP25 for culture cell expression, C178A, which completely abolishes the catalytic activity of USP25 (new **Fig. 6b**). Interestingly, the C178A mutant of the constitutive USP25 dimer does not stabilize tankyrase as the wild-type, displaying levels similar to the negative controls.

Active site mutant (C178A) and endogenous USP25 has been previously analyzed in HEK293 cells by “*Xu et al., USP25 regulates Wnt signaling by controlling the stability of tankyreses. Genes and Dev. 31: 1-12 (2017).*” Alternatively, we have checked the levels of endogenous USP25 in HEK293 cells in a new WB blot with anti-USP25 Ab (new **Figure 6a**). We observed that the levels of endogenous USP25 are fairly small compared to the ectopic USP25 and are equally distributed in all cultures, including the negative culture control (GFP). Our results, together with the previously published in *Genes and Dev. 31: 1-12 (2017)*, indicate that the effect of the endogenous USP25 in the tankyrase stability, compared to the ectopically expressed USP25, is residual.

A new sentence has been added in page 11: “*Also, the inactive USP25 active site mutant C178A produced similar levels of tankyrases as the control cultures, highlighting the role of the ectopic expression of USP25 in the tankyrase stability (Fig. 6b)*”.

- As Tankyrase is a well-established substrate of USp25, it would be easy to purify ubiquitinated tankyrase ((following proteasome inhibition)) and use it as a substrate for assaying enzyme activity of USP25.

Response: As suggested by the reviewer, we have tried to purified endogenous ubiquitinated tankyrase, but with unfruitful results. It is quite hard to observe and purify high amounts of ubiquitinated tankyrase and perform activity assays. Even in the original characterization paper, “*Xu et al., USP25 regulates Wnt signaling by controlling the stability of tankyrases. Genes and Dev. 31:1-12 (2017)*”, the ubiquitinated tankyrases are very hard to observe.

Reviewers' comments:

Reviewer #1 (Remarks to the Author):

This is a revised manuscript by Liu and coworkers on the potential tetrameric structure of USP25. The authors attempted to address the points raised by the two previous reviewers. Although the authors made commendable efforts by including new experiments and revised the manuscript accordingly, there are still major concerns that were not satisfactorily addressed. In fact some of the newly added results raised further questions that are crucial to the validity of the proposed model. Overall the biophysical characterizations in the study were solid. However the experiments on the enzyme kinetics and cellular functions of WT and mutant USP25 lack the necessary rigor (incomplete in several places). These points are listed below and need to be sufficiently addressed for the work to be considered for publication.

1. The authors included a comparison of the USP25NCD with or without the lysine methylation. Notably in gel filtration the features of the methylated form of USP25NCD is different from that of the untreated form. All three peaks (species) observed in the untreated USP25NCD became one peak. There was also a shift in the protein size following the treatment. The authors concluded that the methylation did not affect the integrity of the protein. But it does seem to change the oligomeric state presumably. Why is this? The diUb cleavage assay presented was not of sufficient quality to allow a reliable comparison of the activity of the untreated versus the methylated USP25NCD. A Ub-AMC assay should also be used for a careful comparison of the activity of the two forms of USP25NCD.

2. The binding of IL-loop to the active site of USP25 and the resulting inhibition of the activity is remarkable (somewhat surprising). Given the well defined interactions, the IL-loop alone should also inhibit USP25 activity. Have the authors determined the inhibition of USP25 activity by an IL-loop peptide? If so what was observed? The proposed model of IL-loop binding to the active site of USP25 hence the inhibition of USP25 activity is intriguing (also risky), thus needs stronger experimental supports.

3. In Figure 4e, the kinetics results showed that the double mutant Q322A/E373A showed almost identical activity with the "tetramer", while the single mutant each showed increased activity. This result is very difficult to understand and was not explained in the revised manuscript. It is also worrisome that L271W which should prevent the tetramer formation thus higher activity actually led to significant loss of activity. These results are puzzling. The authors chose to not perform a full kinetic analysis of the point mutants. The lack of k_{cat} and K_m data also casts question on the interpretation of the results.

4. In Figure 6, the authors included a blotting using anti-USP25 antibody comparing the endogenous USP25 level and the ectopically expressed USP25 constructs. The Flag-USP25 FL Δ IL protein level was higher than that of the Flag-USP25 FL, which confirms what was observed for the anti-Flag blotting results. The higher level of Flag-USP25 FL Δ IL protein could be the cause of increased DUB activity, thus higher level of tankyrase level. The authors attributed the increase in tankyrase level to higher activity of Flag-USP25 FL Δ IL, which is not convincing. Have the authors introduce the USP25 FL point mutants such as P521S, F522G, E373A, Q322A, C651F, L271W into the HEK293T cells and determine the level of the tankyrase. The protein expression level should be better controlled at the same level. Also the cells should be treated with proteasome inhibitor to confirm that the level of tankyrase change was due to the ubiquitination status. This is especially important given that the authors could not obtain ubiquitinated tankyrase and test the activity of the WT and mutant USP25 as suggested by the Reviewer 2.

5. In SI Figure S7, the USP25NCD and USP25CD Δ LCC-IL native gel showed several discrete bands. This is hard to understand given that the Δ LCC-IL should lead to the monomeric form based on the model. This raised concern on the analysis.

6. The kinetics result of Δ LCC-IL USP25NCDlong is interesting. However, it is not clear how the two α -helices regulate the activity given that no contact between the two α -helices and the catalytic domain is observed.

7. In page 13, the authors wrote that "a competition analysis (Supplementary Fig. 2b) where high ubiquitin concentrations cannot disrupt the tetramer assembly". The figure numbering is incorrect. It is unexpected that the presence of high concentrations of Ub had no effect on the tetramer structure and activity. This contradicts the kinetics assay results. If the tetramer is so stable and the IL occupies the S1 site, no activity of deubiquitination is expected to be detectable. This casts question of the validity of the proposed inhibition mode of USP25 by the competition of the IL-loop with ubiquitin for the S1 site.

8. To be physiologically relevant the switch between the dimer and tetramer has to be regulated by other cellular events. The author invoked potential phosphorylation of Y454 and the subsequent disruption of the tetrameric structure of USP25. However, Y454 phosphorylation was not detected in the phosphoproteomic studies. Have the authors looked for this modification of Y454 in USP25 expressed in cells and determine what induces the modification. This experiment is highly desirable to establish the physiological relevance of the tetrameric USP25 structure as proposed by the authors.

Reviewer #2 (Remarks to the Author):

The authors have addressed all the major concerns of both reviewers by performing additional experiments and analyses.

The rewritten manuscript is vastly improved as there is a better logical flow.

Minor comment: The new sentence added in page 6, "Apparently,did not affect substantially...". Can the authors please use clear non-ambiguous terms to describe the observed result?

Reviewer #1 (Remarks to the Author):

1. The authors included a comparison of the USP25NCD with or without the lysine methylation. Notably in gel filtration the features of the methylated form of USP25NCD is different from that of the untreated form. All three peaks (species) observed in the untreated USP25NCD became one peak. There was also a shift in the protein size following the treatment. The authors concluded that the methylation did not affect the integrity of the protein. But it does seem to change the oligomeric state presumably. Why is this? The diUb cleavage assay presented was not of sufficient quality to allow a reliable comparison of the activity of the untreated versus the methylated USP25NCD. A Ub-AMC assay should also be used for a careful comparison of the activity of the two forms of USP25NCD.

Response: We are sorry for this misunderstanding; in the initial figure we compared the gel-filtration profile of the purified methylated USP25 tetramer with the original gel-filtration of the USP25 purification, which is composed by dimer and tetramer species (which explains the presence of two major peaks). We have now replaced this figure, and we are now comparing the purified tetramer form of USP25 before and after the methylation protocol (new figure Fig. S2).

Additionally, as requested by the reviewer, we have included the activity assays with Ub-AMC substrate using the methylated and non-methylated form of USP25 tetramer (new figure Fig. S2).

2. The binding of IL-loop to the active site of USP25 and the resulting inhibition of the activity is remarkable (somewhat surprising). Given the well defined interactions, the IL-loop alone should also inhibit USP25 activity. Have the authors determined the inhibition of USP25 activity by an IL-loop peptide? If so what was observed? The proposed model of IL-loop binding to the active site of USP25 hence the inhibition of USP25 activity is intriguing (also risky), thus needs stronger experimental supports.

Response: As suggested by the reviewer, we have designed a peptide corresponding to the IL-loop region that interacts with the USP25 catalytic domain, which is composed by the sequence: HKPFTQSRIPPD. This peptide corresponds to the major interaction region of the IL-loop with USP25 Ub-binding site. Activities assays with Ub-AMC substrate indicate that the peptide does not perturb the activity showed by the USP25 dimer form at two different peptide concentrations (new Supplementary figure Fig. S6E). These results suggest a weak interaction of this peptide alone. There might be two main reasons for this behavior: First, in the tetramer assembly the conformational entropy of the IL-loop is reduced, favoring the interaction with the Ub-binding site. Second, in addition to the IL-loop there is a second interface in the tetramer assembly that is essential for the quaternary assembly, as shown in Supplementary figure Fig. S11.

We have added a new paragraph in page 9: *“Finally, to check a potential competition of the IL-loop with the ubiquitin substrate, USP25 deubiquitinating activities were measured in the presence of a peptide derived from the IL-loop binding sequence (HKPFTQSRIPPD) (Supplemental Fig. 6d). The absence of inhibition produced by the*

IL-loop peptide alone suggests a model in which the structural context in the tetramer assembly is required for the proper binding and thus activity inhibition”.

3. In Figure 4e, the kinetics results showed that the double mutant Q322A/E373A showed almost identical activity with the “tetramer”, while the single mutant each showed increased activity. This result is very difficult to understand and was not explained in the revised manuscript. It is also worrisome that L271W which should prevent the tetramer formation thus higher activity actually led to significant loss of activity. These results are puzzling. The authors chose to not perform a full kinetic analysis of the point mutants. The lack of *k_{cat}* and *K_m* data also casts question on the interpretation of the results.

Response: After careful analysis of the protein quality of the USP25 double mutant (Q322A/E373A), we have decided to remove it from the figure (see new figure Fig. 4 & new supplementary figure Fig. S6). Probably the lack of activity showed by the double mutant was a consequence of the bad protein quality observed after its purification by gel-filtration, which did not behave as the wild-type or the other USP25 single mutants.

In the case of USP25 L271W the results are interesting, this is the only mutant with a loss of activity with the Ub-AMC substrate. Intriguingly this mutant is purified as a tetramer (in contrast to the other mutants, which are dimers, see new supplementary figure Fig. S6A), probably indicating that in L271W point mutant the tetramer assembly is favored by some unexpected interactions. Since this is a non-natural mutation we did not want to explore deeply the reasons for the loss of activity, we believe that it might be a consequence of the tetramer stabilization.

As suggested by the reviewer to shed light on the interpretation of the results we have conducted the kinetic analysis for two USP25 point mutants (E373A and F522G). The results are shown in new Supplementary Fig S6D, and both display *K_m* and *K_{cat}* values similar to the “constitutive” dimer form (USP25NCD Δ IL), indicating that the dimer form of USP25 (active form) is probably favored for these two USP25 point mutants (as also shown by gel filtration analysis) (see new Supplementary Fig S6A).

We have added a new paragraph in page 11: *“Interestingly, two USP25 point mutants that disrupt the tetramer assembly (F522G and E373A), display kinetic values close to the “constitutive” dimer form of USP25 (Supplementary Fig. 6d)”.*

4. In Figure 6, the authors included a blotting using anti-USP25 antibody comparing the endogenous USP25 level and the ectopically expressed USP25 constructs. The Flag-USP25 FL Δ IL protein level was higher than that of the Flag-USP25 FL, which confirms what was observed for the anti-Flag blotting results. The higher level of Flag-USP25 FL Δ IL protein could be the cause of increased DUB activity, thus higher level of tankyrase level. The authors attributed the increase in tankyrase level to higher activity of Flag-USP25 FL Δ IL, which is not convincing. Have the authors introduce the USP25 FL point mutants such as P521S, F522G, E373A, Q322A, C651F, L271W into the HEK293T cells and determine the level of the tankyrase. The protein expression level should be better controlled at the same level. Also the cells should be treated with proteasome inhibitor to confirm that the level of tankyrase change was due to the

ubiquitination status. This is especially important given that the authors could not obtain ubiquitinated tankyrase and test the activity of the WT and mutant USP25 as suggested by the Reviewer 2.

Response: We have attached a figure below to explain the protein expression differences observed in Figure 6. It is true that in the triplicate experiments the ectopic levels of USP25 are higher in the dimer for some unknown expression reasons, however, the differences in the tankyrase levels are still significant (see left panel in the attached figure below). Also, at the right panel, we show another initial figure from one of the triplicate experiments, in which the tankyrase differences are better appreciated.

On the other hand, as suggested by the reviewer, we have analyzed the tankyrase levels in HEK293 cultures of the USP25 point mutants of the IL-loop interface (in which no differences in USP25 expression are expected). The results are shown in figure 6 (see *new Fig. 6*). We hope that the results now better support our hypothesis that the higher deubiquitinating activity of the dimer form of USP25 favors the tankyrase stability.

We have added a new paragraph in page 12: “To confirm the ubiquitin-dependent stability of tankyrase²⁸, cells were treated with cyclohexamide and Bortezomib, which is a potent proteasome inhibitor³⁵. Bortezomib-treated cells revealed higher endogenous tankyrase levels in all cultures analyzed, including the GFP control and the poorly active USP25 L271W point mutant (**Supplementary Fig. 6c**), all confirming the ubiquitin-dependent degradation of tankyrase²⁸ and the role of the deubiquitinating activity of USP25 for the higher stability of tankyrase”.

Finally, to confirm the role of the proteasome-ubiquitin pathway in the half-life regulation of tankyrase, which has already been extensively analyzed in “Xu *et al.*, 2017, USP25 regulates Wnt signaling by controlling the stability of tankyrases. *Genes and Dev.* 31: 1-12”, we have treated our culture cells with cyclohexamide (inhibits novel protein synthesis) and bortezomib (inhibits the 20S-proteasome) to check the stability

of tankyrases. As observed in new figure 6 (see new figure Fig. 6d), bortezomib treatment inhibits the proteasomal degradation of tankyrase in all 4 different cultures.

We have added a new paragraph in page 12: *“Finally, ectopic expression of USP25 point mutants of the IL-loop interface, which weakens the tetramer assembly and impairs the deubiquitinating activity (Fig. 4e, and Supplemental Fig. 6), showed results comparable to the expression of the “constitutive” dimer at different levels (Fig. 6c). Only the L271W point mutant, which already showed a notable loss of activity in vitro, exhibited similar tankyrase levels as in the GFP control cultures. Cyclohexamide treatment of USP25 P521S point mutant yielded similar tankyrase stability as the USP25 dimer (Fig 6d)”*.

5. In SI Figure S7, the USP25NCD and USP25CD Δ LCC-IL native gel showed several discrete bands. This is hard to understand given that the Δ LCC-IL should lead to the monomeric form based on the model. This raised concern on the analysis.

Response: We agree with the reviewer on this point, but our intention was to run in the same native gel the tetramer, dimer and monomer forms of USP25. Unfortunately, the molecular weight of the monomers (around 50 KDa) is very different to the molecular weights of dimers and tetramers (up to 400 KDa), which would probably require a different acrylamide concentration in the gel. This is also more evident when checking the gel migration of the smallest monomer form (USP25 CD Δ LCC-IL), which probably needs different acrylamide concentration to avoid gel diffusion.

Despite the behavior of the monomer USP25 in native gels, we consider that these results are informative and support the analytical gel-filtration and DLS about the oligomeric state of the USP25 truncation constructs.

6. The kinetics result of Δ LCC-IL USP25NCDlong is interesting. However, it is not clear how the two α -helices regulate the activity given that no contact between the two α -helices and the catalytic domain is observed.

Response: In supplementary figure S9 (see new Fig. S9) we have included a zoom-up panel with the contacts between the two helices of the coiled-coil LCC domain and the catalytic domain. They are composed by distinct charged and hydrophobic interactions well observed in the electron density maps. These contacts are distant from the catalytic site, but we speculate that they might stabilize the proper structural conformation required by all the loops that participate in the USP25 catalytic activity.

7. In page 13, the authors wrote that “a competition analysis (Supplementary Fig. 2b) where high ubiquitin concentrations cannot disrupt the tetramer assembly”. The figure numbering is incorrect. It is unexpected that the presence of high concentrations of Ub had no effect on the tetramer structure and activity. This contradicts the kinetics assay results. If the tetramer is so stable and the IL occupies the S1 site, no activity of deubiquitination is expected to be detectable. This casts question of the validity of the proposed inhibition mode of USP25 by the competition of the IL-loop with ubiquitin for the S1 site.

Response: Sorry for this misunderstanding in the figure numbering, we have corrected

it in the version of the manuscript. We agree with the reviewer view on the inhibition mechanism, and we have decided to remove the competition model from the text. Our experiments indicate that ubiquitin neither competes with the Ub-AMC substrate nor disrupts the tetramer assembly (see Fig. S2). Additionally, IL-loop peptide cannot compete with the ubiquitin substrate. Therefore we cannot support a competition inhibition model. We support a two-state model in which USP25 can be either **active (dimer)** or **not active (tetramer)**.

We have been able to measure new kinetic parameters for tetramer, dimer and monomer at high substrate concentrations. The initial reported values were only tentative because activity rates could only be measured up to 10 μM Ub-AMC, but now we have been able to measure activity rates up to 60 μM (which are required to calculate correctly the USP25 K_m values). The final kinetic results display a higher catalytic efficiency for the dimer over the tetramer (see new Fig. 5D).

We have also modified the text accordingly in page 14: *“However, neither high ubiquitin concentrations cannot disrupt the tetramer assembly (**Supplementary Fig. 2D,E**) nor increasing amounts the IL-loop peptide disturbs the USP25 deubiquitinating activity (**Supplementary Fig. 6E**), excluding a direct competition between ubiquitin substrate and the IL-loop peptide”*.

8. To be physiologically relevant the switch between the dimer and tetramer has to be regulated by other cellular events. The author invoked potential phosphorylation of Y454 and the subsequent disruption of the tetrameric structure of USP25. However, Y454 phosphorylation was not detected in the phosphoproteomic studies. Have the authors looked for this modification of Y454 in USP25 expressed in cells and determine what induces the modification. This experiment is highly desirable to establish the physiological relevance of the tetrameric USP25 structure as proposed by the authors.

Response: As invoked by the reviewer we are strongly interested to determine a possible cellular switch that can disrupt the tetramer assembly and activate the deubiquitinase activity of USP25. Phosphorylation might be a plausible switch between the two oligomeric states (as occurs in many other examples of metabolic enzymes). Our efforts in our future project will be to determine a post-translational modification by mass spectrometry of USP25 purified from cells. Unfortunately, our initial efforts in this approach have proved unsuccessful, but we expect to get better results in the very close future.

Y454 residue was a phospho-site described for USP28 (Ross, K.E. *et al.* *iPTMnet: Integrative Bioinformatics for Studying PTM Networks. Protein Bioinformatics: From Protein Modifications and Networks to Proteomics*, 333-353 (2017)). USP28 is highly similar to USP25 (more than 50% identity, see Fig. S1). Our intention with the experiments shown in Fig. S11 was only a proof of concept that a single phosphomimetic modification, Y454E, was able to disrupt the tetramer assembly of USP25.

Reviewer #2 (Remarks to the Author):

The authors have addressed all the major concerns of both reviewers by performing additional experiments and analyses.

The rewritten manuscript is vastly improved as there is a better logical flow.

Minor comment: The new sentence added in page 6, "Apparently,did not affect substantially...". Can the authors please use clear non-ambiguous terms to describe the observed result?

Response: Sorry for that. We have modified the text accordingly in page 6: "Apparently, the lysine methylation reaction did not affect the deubiquitinating activity of USP25 (Supplementary Fig. 2a)".

Reviewers' comments:

Reviewer #1 (Remarks to the Author):

In this revision the authors carried out further experiments to assess the model of USP25 oligomeric structure and its regulation by the tetramer-to-dimer transition in in vitro assays and in cell. Some new experimental results were included and several pieces of previous data were left out. The updated activity results showed that the catalytic activity of USP25NCD tetramer is 2-fold lower than that of USP25NCD Δ IL, while the K_m of USP25NCD Δ IL is 1.5-fold higher than that of the USP25NCD tetramer. However, if the IL loop occupies the S1 ubiquitin-binding site as proposed by the model, a much larger difference in both parameters would be expected. This cannot be sufficiently explained by the very low level of contamination of the dimeric species in the tetramer (Fig. S2). Further, the activity difference in cleaving polyubiquitin by the USP25NCD tetramer, dimer and Δ IL is again very small, in accord with the minor change observed using Ub-AMC.

The cell data on the level of tankyrase also showed small change, based on the quantification in the newly added bar graph accompanying the gel in Figure 6. The P value of the quantification in the bar graph was not clearly defined. What is the P value comparing FL versus FL Δ IL in Figure 6a, which seems to be barely significant.

In Figure 6 CHX treatment provides the rate information of the degradation of the tankyrase in cells. In panel D, the time-dependent decrease of the tankyrase protein level, hence the rate of tankyrase degradation, is not much different for FL and P521S USP25. In panel B, the rate of tankyrase degradation for FL and FL Δ IL USP25 seems not too different, despite that the tankyrase overall protein level is higher for the FL Δ IL construct (however the Flag-USP25 protein level is higher for FL Δ IL USP25 than FL USP25).

Given the small difference in the in vitro deubiquitination and the cellular tankyrase level, an important question remains, how significant and physiologically relevant the proposed quaternary structure of USP25 is. In addition to tankyrase, the authors should assess other known USP25 targeted proteins, such as the ERAD substrates. For tankyrase, other USP25-regulated cellular activities such as the β -catenin-regulated gene expression, β -catenin level and the survival and growth of Wnt-dependent cells need to be assayed in the presence of the FL versus FL Δ IL USP25. These experiments are necessary to ensure that what was observed in the crystal is physiologically relevant in cells.

In authors' response to point 5, the argument of the different size of the USP25 constructs is not particularly convincing, given that the oligomer of USP25NCD Δ LCC-IL has comparable MW to that of USP25NCD Δ LCC. There is a discrepancy between the different methods used in this respect and should not be ignored. Have the authors considered that USP25NCD Δ LCC-IL still retains the ability to form oligomeric structures? The SEC and DSL results seem to tell the same story, but not the non-denaturing gel electrophoresis. If the authors are convinced by the SEC and DSL results, they should consider leaving out the latter to avoid misleading the readers.

In authors' response to point 7, the authors decided to remove the competition model. However, as long as the IL loop occupies the S1 ubiquitin-binding site as in the structural model, this competition is unavoidable. Otherwise, it is very difficult to understand how the dimer to tetramer transition regulates the USP25 activity.

In point 8, Y to E mutation is not a good mimetic for tyrosine phosphorylation.

Reviewer #1 (Remarks to the Author):

In this revision the authors carried out further experiments to assess the model of USP25 oligomeric structure and its regulation by the tetramer-to-dimer transition in vitro assays and in cell. Some new experimental results were included and several pieces of previous data were left out. The updated activity results showed that the catalytic activity of USP25NCD tetramer is 2-fold lower than that of USP25NCD Δ IL, while the K_m of USP25NCD Δ IL is 1.5-fold higher than that of the USP25NCD tetramer. However, if the IL loop occupies the S1 ubiquitin-binding site as proposed by the model, a much larger difference in both parameters would be expected. This cannot be sufficiently explained by the very low level of contamination of the dimeric species in the tetramer (Fig. S2). Further, the activity difference in cleaving polyubiquitin by the USP25NCD tetramer, dimer and Δ IL is again very small, in accord with the minor change observed using Ub-AMC.

Response: In order to check this observation by reviewer 1, we have measured the dimer contamination fraction in three different native gels (**Figure 1a, below**) and compare it to the velocities with Ub-AMC substrate (**Figure 1b, below**).

The results indicate that the % of dimer contamination in the native gels (21 to 31%, depending on the acrylamide concentration) is comparable to the differences in the de-ubiquitinating velocities measured with the Ub-AMC substrate (33 to 37%) (**Figure 1, below**).

These results support our model for USP25, tetramer (not active) vs dimer (active).

Figure 1a Comparison of the % dimer presence in the tetramer fraction from 3 gels, based on densitometry of the depicted native gel bands.

Figure 1b Differences between the initial velocities v_0 at 0,1 μ M substrate concentration (Ub-AMC) and between V_{max} (table from Fig. 5d) from dimer and tetramer samples.

Additionally, since the K_M value for USP25 is high (range 20 - 38 μ M), an accurate measurement of the kinetic parameters would require substrate concentrations around 200 μ M (10 x K_M), which is not achievable. For this reason we think that the linearity displayed in the initial velocity at 0,1 μ M substrate concentration is a more accurate method to compare the activities between dimer and tetramer.

The cell data on the level of tankyrase also showed small change, based on the quantification in the newly added bar graph accompanying the gel in Figure 6. The P value of the quantification in the bar graph was not clearly defined. What is the P value comparing FL versus FL Δ IL in Figure 6a, which seems to be barely significant.

Response: In new figure 6A we have added the statistical analysis required by the reviewer. The P value between FL vs FL Δ IL is significant (included in the figure legend).

In Figure 6 CHX treatment provides the rate information of the degradation of the tankyrase in cells. In panel D, the time-dependent decrease of the tankyrase protein level, hence the rate of tankyrase degradation, is not much different for FL and P521S USP25. In panel B, the rate of tankyrase degradation for FL and FL Δ IL USP25 seems not too different, despite that the tankyrase overall protein level is higher for the FL Δ IL construct (however the Flag-USP25 protein level is higher for FL Δ IL USP25 than FL USP25).

Response: Tankyrase is a direct target of USP25, but its stability in cells may also depend on other factors. However, we believe that the relevant result is the tankyrase stability in the new single point mutants, in which there is no different USP25 expression levels in cells. The results (Fig 6c,d) display significant differences for P521S (and other mutants) after 48h of transient transfection.

Given the small difference in the in vitro deubiquitination and the cellular tankyrase level, an important question remains, how significant and physiologically relevant the proposed quaternary structure of USP25 is. In addition to tankyrase, the authors should assess other known USP25 targeted proteins, such as the ERAD substrates. For tankyrase, other USP25-regulated cellular activities such as the β -catenin-regulated gene expression, β -catenin level and the survival and growth of Wnt-dependent cells need to be assayed in the presence of the FL versus FL Δ IL USP25. These experiments are necessary to ensure that what was observed in the crystal is physiologically relevant in cells.

Response: We would also have liked to analyze other physiological substrates for USP25 to strengthen our mechanism, however, tankyrases are the only bona fine substrates with a complete structural characterization, in which the interaction with the C-terminal domain of USP25 has been mapped. Other downstream substrates in the pathway, such as b-catenin, may also be regulated by other factors and perhaps it would not be so conclusive.

However, our data with tankyrases indicate that:

- The different stability of the endogenous tankyrases between USP25 dimer and tetramer is statistically significant.
- The USP25 active site mutant (C178A) does not stabilize tankyrases, indicating a de-ubiquitinating-activity dependence.
- The stability of endogenous tankyrases by expression of single point mutants of the IL-loop, which disrupt the tetramer assembly, is also statistically significant.

Nevertheless, in agreement to the reviewer's views, we have toned down the physiological significance of our model in the abstract and across the manuscript (please see new version of the manuscript).

In authors' response to point 5, the argument of the different size of the USP25

constructs is not particularly convincing, given that the oligomer of USP25NCD Δ LCC-IL has comparable MW to that of USP25NCD Δ LCC. There is a discrepancy between the different methods used in this respect and should not be ignored. Have the authors considered that USP25NCD Δ LCC-IL still retains the ability to form oligomeric structures? The SEC and DSL results seem to tell the same story, but not the non-denaturing gel electrophoresis. If the authors are convinced by the SEC and DSL results, they should consider leaving out the latter to avoid misleading the readers.

Response: The only reason to generate the monomer truncation of USP25 was to show the role of the new LCC-IL internal domain in the formation of dimer and tetramer (LCC-IL is a novel uncharacterized domain). SEC, DLS and a model from the crystal structure indicate the presence of a monomer. Native gel also points in this direction, despite the presence of other discrete bands that could be contamination, crosslinks ... (difficult to assess in native gels). Nevertheless, we have run again a new native gel (with higher acrylamide concentration) with tetramer, dimer and monomer (**Figure 2**) in which the monomer band is stronger.

Figure 2. Native gel (12,5% acrylamide) of tetramer, dimer and monomer constructs of USP25 NCD. The monomer band is quite strong, despite the presence of discrete bands (contamination?).

Please, notice again the dimer contamination in the tetramer sample (first point of this response).

In authors' response to point 7, the authors decided to remove the competition model. However, as long as the IL loop occupies the S1 ubiquitin-binding site as in the structural model, this competition is unavoidable. Otherwise, it is very difficult to understand how the dimer to tetramer transition regulates the USP25 activity.

Response: Our data indicate that ubiquitin does not interfere in either the de-ubiquitinating activity or in the tetramer assembly of USP25 (**Fig S2**). Our kinetic data, despite the difficult assessment of the K_M (due to its high value), does not support a competitive model. We support a model in which the tetramer assembly of USP25 is not active, and the dimer assembly is active. We speculate that the transition between tetramer and dimer could occur by phosphorylation (or another PTM), not by substrate binding. We indeed show that a single point mutation outside the IL-loop binding region can produce this tetramer to dimer transition (**Fig S11**).

A new sentence in discussion in page 14: "Our working model speculates on a post-translation modification that would promote the transition between tetramer and dimer in the cell" (**Fig. 6**).

In point 8, Y to E mutation is not a good mimetic for tyrosine phosphorylation.

Response: Sorry for that, glutamic acid is not a good phosphomimetic residue for tyrosine, although it is frequently used in the literature.

We have removed a sentence in page 14: "but Y454E, a phosphomimetic tyrosine substitution, results in the formation of ...".